# THRONCAT: metabolic labeling of newly synthesized proteins using a bioorthogonal threonine analog

Bob J. Ignacio[1], Jelmer Dijkstra[2,3], Natalia Mora [4,5], Erik F. J. Slot [4,5], Margot J. van Weijsten[1], Erik Storkebaum [4], Michiel Vermeulen [2,3] & Kimberly M. Bonger [1] ✉

Profiling the nascent cellular proteome and capturing early proteomic changes in response to external stimuli provides valuable insights into cellular physiology. Existing metabolic protein labeling approaches based on bioorthogonal methionine- or puromycin analogs allow for the selective visualization and enrichment of newly synthesized proteins. However, their applications are limited as they often require methionine-free conditions, auxotrophic cells and/or are toxic to cells. Here, we introduce THRONCAT, a threonine-derived non-canonical amino acid tagging method based on the bioorthogonal threonine analog β-ethynylserine (βES) that enables efficient labeling of the nascent proteome in complete growth media within minutes. We use THRONCAT for the visualization and enrichment of nascent proteins in bacteria, mammalian cells and *Drosophila melanogaster*. We profile immediate proteome dynamics of B-cells in response to B-cell receptor activation simply by adding βES to the culture medium, demonstrating the ease-of-use of the method and its potential to address diverse biological questions. In addition, using a *Drosophila* model of Charcot-Marie-Tooth peripheral neuropathy, we show that THRONCAT enables visualization and quantification of relative protein synthesis rates in specific cell types in vivo.

Cells rapidly respond to environmental changes by synthesizing new proteins to ensure proper cell functioning. Characterizing this nascent proteome is important to understand cell physiology, but remains challenging as it requires an approach to distinguish between newly synthesized proteins (NSPs) and the pre-existing proteome. Metabolic protein labeling uses exogenous amino acid- or puromycin analogs, which are incorporated into nascent proteins by the endogenous biosynthetic machinery[1]. These metabolic probes contain bioorthogonal reactive groups for conjugation to fluorescent dyes or affinity tags, enabling selective visualization and enrichment of NSPs (Fig. 1)[2].

The most commonly used metabolic labeling reporters are bioorthogonal methionine analogs azidohomoalanine (AHA) and homopropargylglycine (HPG; Fig. 2a)[3–5], and puromycin analog *O*-propargyl-puromycin (OPP)[6]. OPP labels NSPs within minutes and is easy to use, as labeling can be performed in complete growth media. However, OPP is toxic to cells and yields truncated OPP-polypeptide adducts that are unstable and proteolytically degraded within 1 h[6].

[1]Department of Synthetic Organic Chemistry, Chemical Biology Lab, Radboud University, Heyendaalseweg 135, 6525AJ Nijmegen, the Netherlands. [2]Department of Molecular Biology, Radboud Institute for Molecular Life Sciences, Oncode Institute, Radboud University, Nijmegen, the Netherlands. [3]Division of Molecular Genetics, The Netherlands Cancer Institute, Amsterdam, The Netherlands. [4]Molecular Neurobiology Laboratory, Donders Center for Neuroscience, Donders Institute for Brain, Cognition and Behaviour and Faculty of Science, Radboud University, Nijmegen, the Netherlands. [5]These authors contributed equally: Natalia Mora, Erik F. J. Slot. ✉e-mail: k.bonger@science.ru.nl

**Fig. 1 | Scheme of metabolic protein labeling.** Bioorthogonal analogs of amino acids or puromycin are incorporated biosynthetically into growing polypeptide chains on the ribosome. The metabolic labels are exclusively incorporated into newly synthesized proteins (NSPs), allowing for their selective visualization or enrichment through bioorthogonal ligation to fluorescent dyes or affinity tags, respectively.

Bioorthogonal non-canonical amino acid tagging (BONCAT) with AHA and HPG is less toxic and provides stable labeling in NSPs, but AHA and HPG are poorly incorporated in proteins, and depletion of the intracellular methionine pool is often required to obtain sufficient levels of protein labeling[7].

Above challenges highlight the need for a metabolic protein labeling method that allows fast and efficient incorporation of amino acid analogs in nascent proteins and can be used under native conditions. Threonine analogs 4-fluorothreonine (4-FT)[8] and β-hydroxynorvaline (β-HNV)[9,10] are excellent substrates for the threonyl-tRNA synthetase (ThrRS) and are efficiently incorporated into the nascent proteome of bacteria in complete growth media. We envisioned that the bioorthogonal threonine analog β-ethynylserine (βES; Fig. 2b)[11–13], may also be efficiently incorporated into nascent proteins, facilitating their labeling in cells grown in complete medium.

Here, we introduce THRONCAT, a metabolic labeling method based on <u>thr</u>eonine-derived <u>n</u>on-<u>c</u>anonical <u>a</u>mino acid <u>t</u>agging. We show that threonine analog βES is efficiently incorporated into NSPs, is non-toxic, and allows labeling of the nascent cellular proteome in complete growth medium within minutes. We demonstrate that THRONCAT has high labeling efficiency and compare our method to BONCAT using HPG for the visualization of NSPs in bacteria, mammalian cells, and *Drosophila melanogaster*. Furthermore, we used THRONCAT to profile proteomic changes over time using βES pulse-labeling following B-cell receptor (BCR) stimulation in Ramos B cells, demonstrating the ease-of-use of the method. In addition, by combining THRONCAT with genetic expression of a fluorescent cell marker in motor neurons, we quantified in vivo cell-type-specific changes in protein synthesis rate in a *Drosophila* model of Charcot-Marie-Tooth peripheral neuropathy.

## Results

### βES is efficiently incorporated into the nascent prototrophic *E. coli* proteome

To explore the use of threonine analogs in metabolic labeling experiments, we first devised a stereoselective synthesis route towards βES (Supplementary Fig. 1). Starting from commercially available materials and using an asymmetric aminohydroxylation as the key step in our synthetic route, we obtained βES in four steps in a 22% overall yield.

Next, we explored whether βES is incorporated into the nascent proteome of bacteria. As prototrophic bacteria can synthesize their own pool of methionine, tagging NSPs using methionine derivatives is often challenged by the need for methionine-auxotrophic bacteria and the use of methionine-free growth medium. Indeed, we hardly observed protein labeling using methionine analog HPG in prototrophic *E. coli* BL21 cells that were grown in complete growth medium (Fig. 2c, d) while we confirmed strong incorporation of HPG in

auxotrophic *E. coli* B834 cells in methionine-free growth medium (Fig. 2e). In contrast to HPG, supplementing prototrophic *E. coli* BL21 cells growing in complete medium with βES resulted in strong labeling of the nascent proteome (Fig. 2c, d). The labeling occurred throughout the proteome, in a time-dependent manner (Fig. 2f) and without inhibiting bacterial growth (Fig. 2g). Incorporation of βES was abrogated by the addition of protein synthesis inhibitor chloramphenicol (CAP) or a 50-fold excess of threonine, suggesting that βES is incorporated exclusively into NSPs at positions encoding for threonine (Fig. 2g, h). Interestingly, looking at the relative labeling intensities of individual protein bands in the SDS-PAGE gels, we observed a clear difference between βES- and HPG-labeled *E. coli* lysate, possibly because of varying numbers of threonine- and methionine residues in individual proteins (Fig. 2e).

### βES allows for fast visualization of mammalian NSPs

Encouraged by the observation that βES is efficiently incorporated in NSPs of prototrophic *E. coli* in complete medium, we next explored the efficiency of βES labeling of nascent proteins in mammalian cells. We treated HeLa cells for 1 h with βES in complete growth medium and conjugated incorporated βES to a Cy5-azide fluorophore. Fluorescence analysis of HeLa cells by flow cytometry and in-gel fluorescence revealed efficient and concentration-dependent incorporation of βES (Fig. 3a; Supplementary Fig. 2). Co-incubation with protein synthesis inhibitor cycloheximide (CHX, Fig. 3b) or excess threonine (Supplementary Fig. 3) drastically reduced fluorescence, confirming incorporation of βES in newly synthesized proteins at the position of threonine. In complete medium, the fluorescence signal was discernible from background using the lowest (4 μM) βES concentration tested and increased dose-dependently, giving a ~200-fold increase in signal over background at the highest concentration tested (4 mM βES; Fig. 3a). In contrast, 1 h incubation of HeLa cells with 4 mM HPG in complete medium yielded only minimal HPG incorporation in NSPs (Fig. 3a, b). Interestingly, while clear differences in labeling levels were obtained for HPG and βES in minimal medium at lower concentrations of analog (4 μM, Supplementary Fig. 4), similar levels of labeling were obtained at high concentration of analog (e.g., 4 mM, Fig. 3b), suggesting that a saturated level of labeling was reached under these experimental conditions. Importantly, we could also obtain a 200-fold signal over background using a 1000-fold lower (4 μM) βES concentration when cells were grown in threonine-free medium (Supplementary Fig. 4).

Fluorescence microscopy and flow cytometry revealed that labeling with 4 mM βES in complete medium resulted in strong fluorescent labeling of the nascent HeLa proteome after Cy5-azide conjugation within minutes (Fig. 3c, d, Supplementary Fig. 5). Cycloheximide treatment greatly reduced the fluorescent signal from βES labeling, confirming the specific incorporation of βES into NSPs (Fig. 3c, Supplementary Fig. 5). Labeled NSPs were distributed throughout the cell and the strongest fluorescence was observed in the nucleoli, which is consistent with the rapid accumulation of ribosomal proteins in the nucleolus[14] and with previous results obtained with HPG and OPP labeling[4,6].

Intrigued by the strong fluorescent labeling attained with βES in HeLa cells, we determined the incorporation rate of βES relative to threonine using mass spectrometry-based proteomics. We incubated HeLa cells simultaneously with 1 mM βES and 1 mM heavy threonine (Thr$_5$) for 72 h in threonine-free culture medium, lysed the cells, and subjected digested peptides to LC-MS/MS analysis. From the ~10,000 labeled peptides we found 142 peptides containing βES indicating a minimal labeling efficiency of 1.4% (Supplementary Fig. 7a). Paired comparison between peak intensities of the βES-/Thr$_5$-modified peptides (Supplementary Fig. 7a) showed βES replaced 1 in 40.8 threonine residues on average, indicating a relative incorporation rate of ~1:40 (βES:threonine) (Supplementary Fig. 7b, c, Supplementary Data 1). Because the incorporation rate of HPG relative to methionine has been

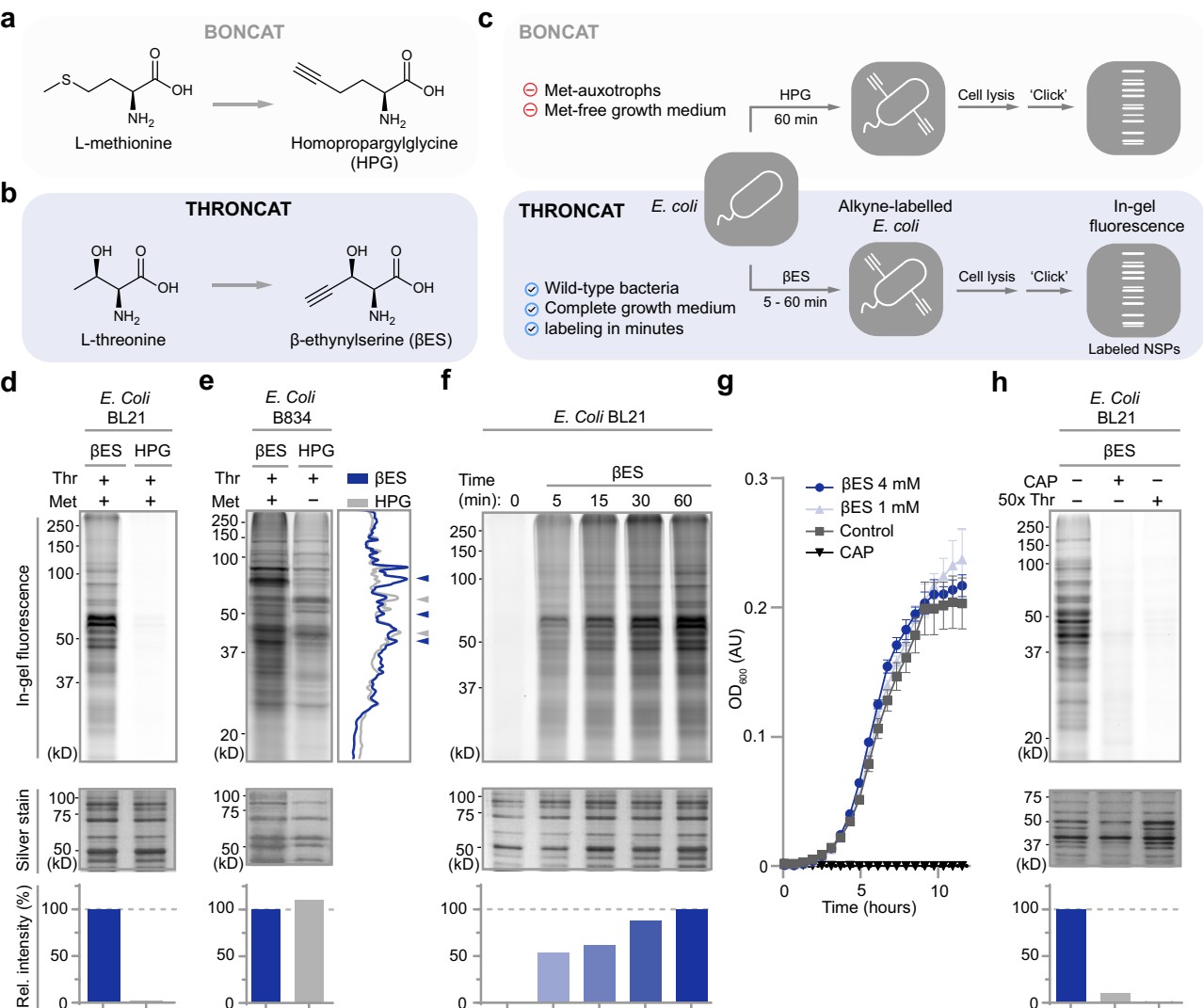

**Fig. 2 | Characterization of βES labeling in bacteria. a** Structures of L-methionine and its analog homopropargylglycine (HPG) used in BONCAT. **b** Structures of L-threonine and its analog β-ethynylserine (βES) used in THRONCAT. **c** Scheme of the BONCAT and THRONCAT workflows for NSP labeling in *E. coli* and their associated drawbacks and benefits. **d–f**, **h** In-gel visualization of βES or HPG incorporated into *E. coli*. Incorporated βES or HPG was conjugated to Cy5-azide using a copper-catalyzed azide-alkyne cycloaddition reaction and visualized through in-gel fluorescence scanning. Silver stain panels show total protein. Bar charts show the relative intensity of in-gel fluorescence normalized to silver stain intensity. The experiments were performed in biological triplicates with similar results. Representative gels are shown here. **d** Comparison of βES and HPG incorporation into E. coli BL21 lysate after 1 h incubation with 4 mM βES or 4 mM HPG in LB medium. A relative intensity bar chart is normalized to βES signal, which is set to 100% **e** Detection of βES and HPG in methionine auxotrophic E. coli B834 lysate after 1 h incubation with 4 mM βES or 4 mM HPG in methionine-free medium. Line scan of gel lanes shows differences in labeling intensities between corresponding bands in

βES (blue line) and HPG (gray line) treated samples. Arrows indicate selected bands showing strong relative labeling in βES (blue arrows) and HPG (gray arrows) treated samples. Relative intensity bar chart is normalized to βES signal, which is set to 100%. Met, Methionine. **f** βES incorporation in *E. coli* BL21 over time. *E. coli* was incubated with 4 mM βES in LB medium for the indicated durations. Relative intensity bar chart is normalized to 'βES 60 min', which is set to 100%. **g** Growth curve of E. coli BL21 incubated with 1 mM or 4 mM βES. Bacterial growth was monitored by OD600 absorbance. Untreated *E. coli* (control) and E. coli treated with 34 µg/mL chloramphenicol were included as positive and negative controls, respectively. $OD_{600}$, optical density at 600 nm; AU, arbitrary units. Data are presented as mean values, error bars represent s.d., Sample size is $n = 6$. **h** βES detection in E. coli BL21 after 1 h incubation with 1 mM βES in LB medium. No βES incorporation is detected upon co-incubation with chloramphenicol (CAP) or a 50-fold excess of threonine. Relative intensity bar chart is normalized to 'βES without CAP or Thr' signal, which is set to 100%. CAP chloramphenicol, Thr L-threonine, kD Kilodalton. Source data are provided as a Source Data file.

determined to be considerably lower (1:500)[4,15,16], we hypothesized that βES incorporation should be less affected by competition from threonine than HPG incorporation by competition from methionine. Indeed, competition of threonine decreases βES incorporation in HeLa cells, but methionine strongly inhibits HPG incorporation at lower concentrations (Fig. 3e). Together, these data show that robust βES labeling in complete medium is enabled by its relatively high incorporation rate when compared to HPG.

Finally, we assessed whether cell fitness was affected by the incorporation of βES over longer time periods. We incubated HeLa

cells up to 24 h with βES in complete medium and observed a steady increase of βES incorporation as quantified by flow cytometry (Supplementary Fig. 8). Importantly, we did not observe a reduction in cell viability at 24 h in the presence of 0.4–4 mM βES as measured by propidium iodide exclusion (Supplementary Fig. 9). Moreover, proteomic analysis showed that labeling with 1 mM βES for 5 h did not induce a cellular stress response (Supplementary Fig. 10). In addition, we did not observe an effect on cell proliferation of HeLa cells exposed to up to 0.4 mM βES for 24 h. Incubation of cells with the highest βES concentration,

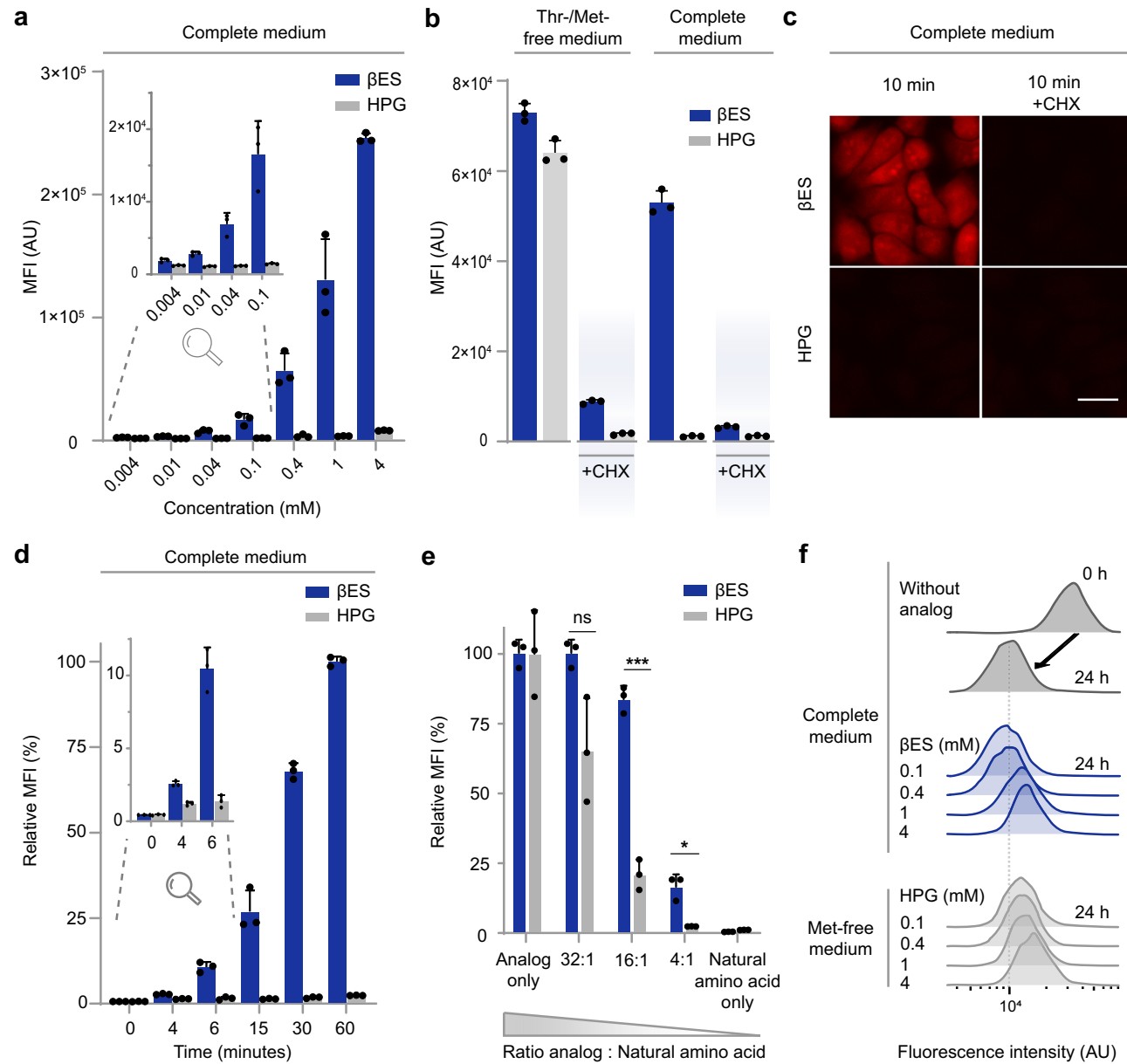

**Fig. 3 | Visualization of NSPs in mammalian cells using THRONCAT.**
**a**–**e** Visualization and quantification of incorporated analogs was enabled by conjugation to Cy5-azide. **a** Flow cytometry quantification of βES or HPG incorporation into HeLa cells. HeLa cells were incubated for 1 h with the indicated concentration of analogs in complete medium. Top-left inset shows zoom of 0.004–0.1 mM concentrations. MFI, mean fluorescence intensity; AU, arbitrary units. Error bars represent s.d. Sample size is $n = 3$. **b** Flow cytometry quantification of βES or HPG incorporation into HeLa cells after 1 h incubation with analog. HeLa cells were labeled with 4 mM βES or 4 mM HPG in threonine-free or methionine-free medium, respectively, or in complete medium. Where indicated, cycloheximide (CHX) was added as a control. Error bars represent s.d. Sample size is $n = 3$. **c** Representative fluorescent images of HeLa cells treated for 10 min with 4 mM βES or 4 mM HPG in complete medium, with or without cycloheximide, followed by conjugation to Cy5-azide (red). Scale bar, 20 μm. **d** Flow cytometry time course analysis of βES and HPG incorporation into HeLa proteomes. HeLa cells were incubated with 4 mM analog in complete medium for the indicated duration. Top-left inset shows zoom of time

points 0–6 min. Signals are normalized to that of βES at 60 min, which is set to 100%. Error bars represent s.d. Sample size is $n = 3$. **e** Flow cytometry quantification of the inhibitory effect of the natural amino acids, threonine or methionine, on the incorporation of their analog, βES or HPG, respectively. HeLa cells were incubated with 50 μM analog and increasing ratios of threonine or methionine, respectively. For both analogs, signals are normalized to their respective signals for 'analog only', which are set to 100%. ns, not significant. ***$P = 0.0001$ and *$P = 0.0305$; determined by an unpaired two-tailed t test. Error bars represent s.d. Sample size is $n = 3$. **f** Flow cytometry proliferation assay of HeLa cells. HeLa cells were labeled with Celltrace Violet and incubated without analog (control, dark gray) or with indicated concentrations of βES (blue) or HPG (light gray) for 24 h. Black arrow indicates decrease in Celltrace fluorescence in control cells. Dotted line indicates final fluorescent signal obtained from control cells. Experiment was performed in triplicate (see Supplementary Fig. 6 for statistics and exact $p$ values), data presented in **f** is from a representative replicate. Source data are provided as a Source Data file.

however, resulted in a slight, but statistically significant decrease in cell proliferation compared to control cells. Impaired proliferation was also observed for HeLa cells treated with lower concentrations of HPG in methionine-free medium (Fig. 3f, Supplementary Fig. 6).

## THRONCAT enables identification and quantification of the dynamic proteome

Next, we used THRONCAT for the enrichment and detection of NSPs of HeLa cells and compared the identified proteins with those observed when using BONCAT in a mass spectrometry-based

proteomics setup. To this end, we incubated HeLa cells in a complete medium for 5 h with 4 mM βES in three biological replicates, enriched NSPs and subjected digested peptides to LC-MS/MS analysis (Fig. 4a). After subtraction of 501 background proteins (Supplementary Fig. 11), THRONCAT identified 3073 unique NSPs in HeLa cells (Fig. 4b, Supplementary Data 2). Gene ontology analysis on identified NSPs shows comprehensive coverage of major cellular compartments and shows a similar subcellular distribution as NSPs identified with BONCAT, confirming broad incorporation of βES throughout the proteome (Supplementary Fig. 12). Comparison of the individual replicates shows that ~83% of NSPs are found in all three replicates and ~90% in at least 2 replicates, indicating good experimental reproducibility (Supplementary Fig. 13). Using an original Python script (Supplementary Software), we determined that the average threonine content of identified proteins is 5.20%, which is identical to the threonine content of the human proteome, showing that THRONCAT enrichment is not biased towards threonine-rich proteins. Also, compared to a dataset with HeLa protein half-life values[17] and to BONCAT-enriched NSPs, THRONCAT-enriched NSPs have similar median protein stability (Supplementary Fig. 14), indicating that THRONCAT enrichment is not excessively biased towards proteins with either high or low turnover.

Labeling with a four-fold lower concentration of βES (1 mM), resulted in the identification of 2740 NSPs after subtraction of background proteins, which is only a ~10% reduction (Fig. 4b). Importantly, using BONCAT−4 mM HPG in methionine-free conditions−we identified a similar number of HeLa NSPs as with 4 mM βES in complete medium (Fig. 4b). Overlap between NSPs identified by THRONCAT and BONCAT was large (81%) and only a fraction of NSPs were identified solely by THRONCAT (9.4%) or BONCAT (9.6%) (Fig. 4c, Supplementary Data 3). Interestingly, the THRONCAT-only fraction was enriched in proteins low in methionine content and conversely, the BONCAT-only fraction was enriched with proteins low in threonine content (Supplementary Fig. 15, Supplementary Data 3).

Because of its fast kinetics and easy workflow, we envisioned that THRONCAT may enable capturing immediate protein abundance dynamics following cell stimulation by pulse-labeling cells at different time intervals. We stimulated Ramos B cells with anti-IgG and pulse-labeled cells for 1 h at different time points with 1 mM βES as well as $^{13}C_6,^{15}N_2$-L-lysine ($Lys_8$), which provided a unique internal marker for NSPs. Measuring 3 biological replicates, we pulse-labeled 0−1 h, 1−2 h, and 3−4 h after B-cell stimulation and included a pre-stimulation control representing the steady state proteome (Fig. 4d).

As expected, the large majority of detected peptides contained $Lys_8$ as opposed to $Lys_0$, indicating that THRONCAT enriches for NSPs with high specificity (Supplementary Fig. 16). Interestingly, we detected 656 NSPs with very diverse expression profiles over the course of B-cell stimulation. We found that of the 579 proteins identified during steady-state conditions, 169 (~32%) were not detected 0−1 h after anti-IgG stimulation, of which 114 (~20%) were detected again at 1−2 h or 3−4 h after stimulation, suggesting that their expression is only temporarily down-regulated upon B-cell activation (Supplementary Data 4; Supplementary Fig. 17, 18). Moreover, focusing on NSPs detected in all replicates of all time points, we found 27 differentially expressed proteins ($P < 0.05$, fold change >1.5) (Fig. 4e). For instance, de novo expression of FCRLA− an intracellular B-cell protein with Fc receptor binding properties[18]−reached peak levels 0−1 h following B-cell activation, (Fig. 4f) while de novo expression of sialyltransferase STGAL6[19] increased continuously for 4 h (Fig. 4g). Similarly, we observed initial up-regulation of the synthesis of transcription factor MEF2B[20] followed by down-regulation after 2 h (Fig. 4h), while de novo

expression of structural protein ezrin[21] decreased continuously from 0−4 h (Fig. 4i).

## THRONCAT allows in vivo analysis of protein synthesis in *Drosophila melanogaster*

We next explored whether βES is incorporated into NSPs in *Drosophila melanogaster*, a living and behaving model organism with a rich genetic tool kit. We fed late second instar/early third instar *Drosophila* larvae that selectively express a membrane-tethered green fluorescent protein in motor neurons (*OK371-GAL4 > UAS-mCD8::GFP*) for 48 h on *Drosophila* medium containing 4 mM βES (Fig. 5a). Following labeling, NSPs in the larval central nervous system (CNS) and body wall muscles were conjugated to TAMRA-azide (TAMRA-N₃). Confocal microscopy revealed robust THRONCAT labeling in both the ventral nerve cord (VNC, equivalent to the vertebrate spinal cord) and body wall muscles of larvae incubated with 4 mM βES compared to an untreated control (Fig. 5b−d, Supplementary. Fig 19). Because of the relevance of protein synthesis (defects) in motor neurons for neuromuscular diseases, we decided to focus our subsequent studies on motor neurons in the VNC. We observed βES incorporation in motor neurons (Fig. 5c), which was strongly concentration-dependent, decreasing ~90% at a 10-fold lower concentration of βES (0.4 mM) (Fig. 5e).

Then, we investigated the in vivo labeling kinetics of βES in comparison to HPG by incubating larvae for different durations with either analog. Signal intensity increased significantly over time for both analogs but increased faster for βES and reached a higher maximum level (Supplementary Table 1, Fig. 5f). Notably, 16 h exposure to βES resulted in a comparable signal intensity as 48 h exposure to HPG, illustrating that THRONCAT can be used to efficiently label NSPs in vivo within shorter time frames. We further compared in vivo THRONCAT to the previously established in vivo cell-type-specific MetRS$^{L262G}$-ANL FUNCAT approach[22,23]. The latter method involves the expression of a MetRS$^{L262G}$ transgene in the cell type of interest using the UAS-GAL4 system[24], followed by feeding of larvae or flies with the methionine analog azidonorleucine (ANL). In contrast to endogenous MetRS, MetRS$^{L262G}$ is able to aminoacylate tRNA$^{Met}$ with ANL, resulting in ANL incorporation in NSPs selectively in cells expressing MetRS$^{L262G}$. We expressed MetRS$^{L262G}$ in motor neurons (*OK371-GAL4*) and compared the in vivo labeling kinetics of THRONCAT and cell-type-specific FUNCAT in motor neurons. βES-mediated labeling was substantially more intense at each time point evaluated and increased much faster as compared to MetRS$^{L262G}$-ANL-mediated labeling (Fig. 5g). Strikingly, MetRS$^{L262G}$-ANL FUNCAT labeling was not higher than background after 4 and 8 h of labeling, and specific labeling of motor neurons was only detected from 16 h of labeling onwards (0 h: $3.61 \pm 0.12$ versus 16 h: $7.69 \pm 1.37$; $n = 8$−9; $P = 0.0003$ by Mann-Whitney test). In contrast, already after 4 h of βES labeling the THRONCAT signal was significantly higher than in controls (0 h: $3.62 \pm 0.40$ versus 4 h: $11.65 \pm 1.00$; $n = 8$; $P = 0.0002$ by Mann-Whitney test). Thus, THRONCAT labeling is more efficient than existing metabolic labeling techniques in *Drosophila*.

Next, we determined whether in vivo THRONCAT can be used to quantify changes in protein synthesis rates. Using in vivo cell-type-specific MetRS$^{L262G}$-ANL FUNCAT[22], we previously showed that expression of human glycyl-tRNA synthetase (GlyRS) carrying mutations that cause Charcot-Marie-Tooth (CMT) peripheral neuropathy reduce global protein synthesis in *Drosophila* motor and sensory neurons by ~30 to 60%, depending on the specific mutation, cell type, and ANL labeling time[23,25]. This inhibition of protein synthesis is attributable to the sequestration of tRNA$^{Gly}$ by CMT-mutant GlyRS, resulting in insufficient supply of glycyl-tRNA$^{Gly}$ to the ribosome and ribosome stalling on glycine codons[25]. Thus, we generated *Drosophila* lines that co-express G240R mutant GlyRS and mCD8::GFP in motor neurons (*OK371-GAL4*), and exposed larvae to 4 mM βES for 48 h.

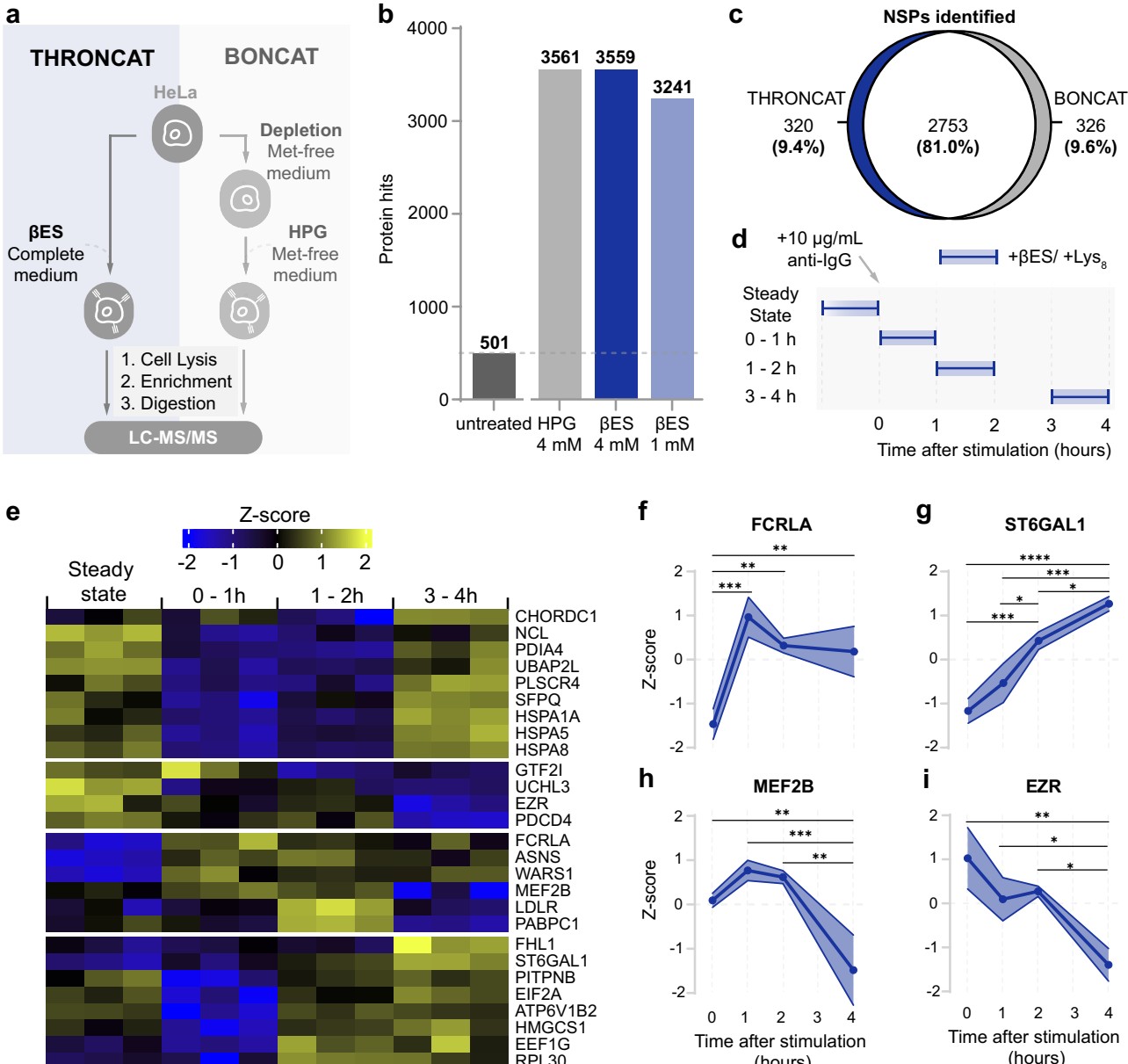

**Fig. 4 | Mass spectrometric analysis of the nascent mammalian proteome with THRONCAT. a** Scheme showing THRONCAT and BONCAT workflows for proteomic analysis of NSPs. NSP, newly synthesized protein; LC-MS/MS, Liquid chromatography-tandem mass spectrometry. **b** Number of HeLa protein hits identified by THRONCAT and BONCAT. HeLa cells were left untreated or incubated for 5 h with 4 mM βES in complete medium (THRONCAT) or starved in methionine-free medium for 30 min and then incubated with 4 mM HPG in methionine-free medium for 5 h (BONCAT). Cells were lysed and NSPs were selectively enriched, digested, and their peptides subjected to LC-MS/MS analysis. Untreated HeLa cells were used as a control. Number of protein hits are presented in bold above each bar. Experiment was performed with three biological replicates per condition. **c** Venn diagram showing overlap between NSPs identified with THRONCAT and BONCAT. NSPs are defined as proteins that are identified in all replicates of treated (THRONCAT/BONCAT) conditions and either absent in the untreated condition or significantly enriched compared to the untreated condition (FC > 1.5, $p$.adj <0.05). False discovery rate was used to correct for multiple testing, with a $q$ value threshold of 0.05. **d** Schematic overview of time course pulse-labeling of stimulated Ramos B cells using THRONCAT. Blue boxes indicate pulse-labeling time windows

during which Ramos cells were exposed to 1 mM βES and Lys$_8$. Ramos cell activation was initiated by the addition of 10 μg/mL anti-IgG at $t = 0$. Experiment was performed with three biological replicates per time window. Anti-IgG, Anti-immunoglobin G antibody; lys$_8$, $^{13}C_6,^{15}N_2$-L-lysine. **e** Heatmap of differentially expressed proteins after Ramos cell activation. Enriched NSPs were quantified by label-free quantification and their abundances normalized by $z$ score. Only identified proteins that were present in all replicates of all-time windows and had an adjusted $p < 0.05$ (two-sided) and fold change >1.5 are included. False discovery rate was used to correct for multiple testing, with a $q$ value threshold of 0.05. **f–i** Expression profiles of FCRLA, ST6GAL1, MEF2B, and EZRIN constructed with z scores from the heatmap shown in **e**. Significance determined by one-way ANOVA, with Bonferroni correction for multiple testing. **f** FCRLA, Fc receptor-like A. From left to right, ***$P = 0.0006$, **$P = 0.0033$, **$P = 0.0054$. **g** ST6GAL1, B-galactoside alpha-2,6-disialyltransferase 1. From left to right, ****$P < 0.0001$, ***$P = 0.0007$, *$P = 0.0162$, ***$P = 0.0003$, *$P = 0.0324$. **h** MEF2B, Myocyte enhancer binding factor 2B. From left to right, **$P = 0.0083$, ***$P = 0.0009$, **$P = 0.0014$. **i** EZR, Ezrin. From left to right **$P = 0.0027$, *$P = 0.0258$, *$P = 0.0131$. **f–i** Error bars represent s.d. Sample size is $n = 3$. Source data are provided as a Source Data file.

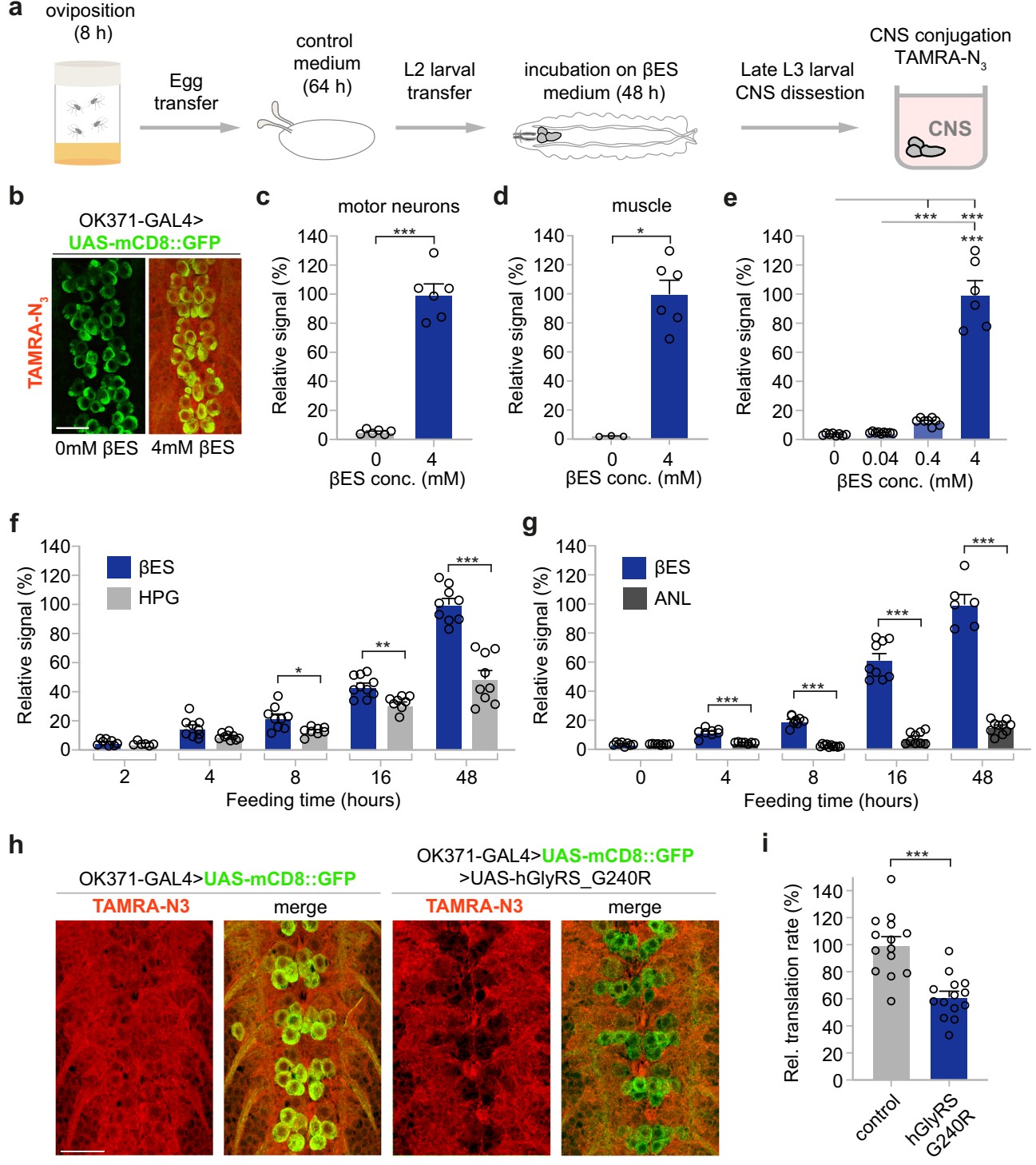

In vivo THRONCAT revealed that in motor neurons (identified as mCD8::GFP expressing cells in the VNC) the labeling intensity was reduced by ~40% upon expression of GlyRS_G240R, as compared to motor neurons expressing mCD8::GFP alone (Fig. 5h, i). This result demonstrates that THRONCAT can be used to detect and quantify cell-type-specific changes in protein synthesis rates in *Drosophila* when combined with fluorescent cell markers.

Finally, we determined whether prolonged THRONCAT labeling affected *Drosophila* fitness or development. Administration of 4 mM βES to larvae during a 48 h time frame did neither affect neuromuscular junction morphology or synapse length, nor motor neuron cell body area, but modestly reduced larval growth and induced a slight developmental delay, while administration of 4 mM βES for 48 h to adult flies did not affect motor performance (Supplementary Figs. 20–22; Supplementary Discussion 1).

## Discussion

Insight into proteome dynamics and regulation is crucial to our understanding of cellular physiology and biochemistry. Metabolic labeling experiments using amino acid derivatives that contain a bioorthogonal handle allow researchers to study, isolate and identify specific subsets of the proteome providing valuable insights in, for example, cellular responses to extracellular stimuli or intercellular exchange of proteins. In this work we have used the non-canonical

**Fig. 5 | THRONCAT allows in vivo analysis of protein synthesis in Drosophila melanogaster. a** Schematic overview of THRONCAT workflow in Drosophila larvae. CNS, central nervous system; TAMRA-N$_3$, 5-carboxytetramethylrhodamine-azide. **b** Representative images of in vivo THRONCAT in Drosophila larval CNS selectively expressing membrane-tethered GFP in motor neurons (OK371-GAL4 > UAS-mCD8::GFP). Larvae were exposed to medium containing either 0 mM or 4 mM βES, followed by conjugation to TAMRA-N$_3$. Merged images of the green (GFP) and red (TAMRA) channels are shown. Scale bar: 20 μm. **c, d** Relative fluorescent signal intensity in motor neurons (**c**) and muscle (**d**) of larvae exposed to 0 mM or 4 mM βES for 48 h and subjected to TAMRA-N$_3$ conjugation. Signals are normalized to that of 4 mM βES, which is set to 100%. **c** $n = 6$ and **d** $n = 3$ (0 mM) or 6 (4 mM) larvae per concentration; ***$P = 0.0022$ (**c**), *$P = 0.0238$ (**d**) by two-tailed Mann–Whitney test. Error bars represent SEM. **e** Relative fluorescent signal intensity in larvae exposed to 0, 0.04, 0.4, or 4 mM βES for 48 h, after conjugation to TAMRA-N$_3$. Signals are normalized to that of 4 mM βES, which is set to 100%. $n = 8$ (0 mM), 9 (0.04 mM), 8 (0.4 mM) or 6 (4 mM) larvae per concentration; ***$P = 0.0019$ (0 vs. 0.4 mM), ***$P < 0.0001$ (0 vs. 4 mM), ***$P = 0.0048$ (0.04 vs 4 mM) by Kruskall–Wallis test. Error bars represent SEM. **f** Relative fluorescent signal intensity after conjugation to TAMRA-N$_3$ in larvae exposed to 4 mM βES or 4 mM HPG for 2 h, 4 h, 8 h, 16 h, or 48 h. $n = 6$–10 larvae per treatment group and time point, For βES from low to high concentrations, $n = 8, 9, 9, 10$, or 10. For HPG from low to high concentrations, $n = 6, 9, 8, 9$ or 9; *$P = 0.0395$ (8 h), **$P = 0.005$ (16 h); ***$P < 0.0005$

(48 h) by unpaired two-tailed $t$ test (2 h, 16 h, 48 h) or two-tailed Mann–Whitney test (4 h, 8 h) per time point, with Bonferroni correction for multiple testing. See Supplementary Table 1 for exact p-values. Error bars represent SEM. **g** Relative fluorescent signal intensity after conjugation to TAMRA-N$_3$ in larvae exposed to 4 mM βES versus OK371-GAL4 > UAS-MetRS$^{L262G}$ larvae conjugated with TAMRA-alkyne after exposure to 4 mM ANL for 0 h, 4 h, 8 h, 16 h, or 48 h. $n = 6$–10 larvae per treatment group and time point, For βES from low to high concentrations, $n = 8, 8, 9, 9$, or 6. For ANL from low to high concentrations, $n = 8, 7, 10, 9$ or 10; ***$P = 0.0015$ (4 h), ***$P < 0.0005$ (8 h), ***$P < 0.0005$ (16 h), ***$P < 0.001$ (48 h) by two-tailed Mann–Whitney test per time point, with Bonferroni correction for multiple testing. Error bars represent SEM. **h** Representative images of THRONCAT in Drosophila larvae selectively expressing membrane-tethered GFP in motor neurons (OK371-GAL4 > UAS-mCD8::GFP), with or without co-expression of G240R mutant human glycyl-tRNA synthetase (UAS-hGlyRS_G240R). Larvae were exposed to 4 mM βES for 48 h, followed by conjugation to TAMRA-N$_3$. The red (TAMRA) channel alone and merged images of the green (GFP) and red channels are shown. Scale bar: 20 μm. **i** Relative translation rate in motor neurons (OK371-GAL4) as determined by THRONCAT in larvae expressing hGlyRS_G240R versus driver-only control. Signals are normalized to that of control, which is set to 100%. $n = 14$ larvae per genotype; ***$P < 0.0001$ by unpaired two-tailed $t$ test. Error bars represent SEM. Source data are provided as a Source Data file.

amino acid βES to establish the metabolic protein labeling method THRONCAT, that is based on an alkyne-bearing threonine analog. βES is incorporated biosynthetically into nascent proteins in the position of threonine, presumably through ThrRS catalyzed aminoacylation of tRNA$^{Thr}$. Incorporation of βES into NSPs is efficient in complete growth medium, in contrast to metabolic labeling approaches based on methionine analogs HPG and AHA that often require depletion of the intracellular methionine pool to allow for strong protein labeling. The higher relative incorporation rate of βES compared to HPG (-12.5-fold higher), combined with the fact that threonine is more abundant than methionine in the human proteome (5.2% vs. 2.5%, respectively) and more solvent exposed in proteins[26] likely contribute to the enhanced labeling of NSPs when using THRONCAT in complete medium. Even though a few percent of all threonine sites are replaced with βES in complete medium, we foresee that the incorporation level is sufficient to cover NSPs throughout the proteome. Indeed, after labeling HeLa cells for 5 h, we identified more than 3000 NSPs after enrichment, which is comparable to that observed when using HPG in a methionine-depleted medium (Fig. 4b and Supplementary Fig. 12).

Methionine depletion, although not essential for all applications of metabolic labeling with methionine analogs[27–30], has been reported to reduce histone synthesis and biomolecule methylation, and to impede cell cycle progression[31,32]. Indeed, we also observed a reduced proliferation rate of HeLa cells under methionine-depleted conditions at all measured HPG concentrations. We expect THRONCAT to be especially suited for NSP labeling in fastidious cell types, such as primary cells, that may need specialized media to support their growth. Moreover, THRONCAT enables facile NSP labeling in wild-type strains of bacteria, thereby precluding the need for auxotrophic strains and increasing the scope of bacteria amenable to metabolic protein labeling. Notably, using THRONCAT in *E. coli* we did not observe the impaired bacterial growth that has previously been observed with BONCAT[33].

Although our proteomic experiments in HeLa cells show that THRONCAT provides thorough NSP enrichment, a fraction of proteins was enriched exclusively by BONCAT. In this context, we note that 2.1% of protein sequences in the human proteome do not contain threonine residues and are therefore not amenable to THRONCAT labeling. Conversely, 2.7% of protein sequences does not contain methionine and 9.1% of proteins contain a single N-terminal methionine residue that may be post-translationally removed.

In experiments where exhaustive identification of NSPs is crucial, a combination of THRONCAT and BONCAT may be used to increase proteomic coverage. We believe THRONCAT will find wide application for analysis of proteomic changes such as cellular activation and differentiation, by facilitating simple and fast pulse labeling of NSPs. Although label-free quantification (LFQ) provides quantification of NSPs in a simple workflow[34], we suggest that THRONCAT is compatible with stable isotope labeling by amino acids in culture (SILAC) or tandem mass tags (TMT) for more accurate quantification of protein expression.

Using a *Drosophila* model of CMT peripheral neuropathy, we showed that THRONCAT enables in vivo quantification of protein synthesis rates. Although βES incorporation is not inherently cell-type specific, a combination of THRONCAT and fluorescent marker expression in a cell type of interest enabled cell-type specific visualization of protein synthesis. THRONCAT provided stronger and faster fluorescent NSP labeling than either HPG- or MetRS$^{L262G}$-ANL FUNCAT in vivo and has the advantage that laborious cell type-specific expression of a MetRS$^{L262G}$ transgene is not required. As such, we foresee broad application of the combination of THRONCAT and cell-type-specific fluorescent reporters for the cell-type-specific visualization and quantification of protein synthesis in (co)-cultured cells or whole organisms.

Although we observed no acute toxicity in mammalian cells or in vivo in *Drosophila* while using THRONCAT, we did observe a reduction in proliferation rate in HeLa cells and a modest growth impairment and developmental delay in *Drosophila* larvae when using the highest βES concentration (4 mM) for long incubation periods. Growth rates of E. coli were, however, unaffected by THRONCAT labeling, even at its highest concentration. Similarly, 48 h βES treatment did not affect motor neuron cell body area, neuromuscular junction morphology and synapse length, and adult motor performance in *Drosophila*. When applying THRONCAT for longer durations (≥24 h), optimization of the βES concentration may be prudent to prevent detrimental effects on proliferation or growth.

In summary, we introduced THRONCAT, a non-canonical amino acid tagging method based on bioorthogonal threonine analog βES, which enables efficient labeling of newly synthesized proteins in complete growth media within minutes. Besides the ease of use, we foresee that the efficient incorporation of βES in NSPs creates opportunities to examine cell responses and mechanisms in models that are currently challenging to study.

## Methods

### Reagents and chemical synthesis

All commercial chemicals and solvents were purchased from Sigma Aldrich, Fluorochem, TCI, or Fisher Scientific and used without further purification. The identity and purity of each product were characterized by $^1$H NMR, $^{13}$C NMR, HRMS, TLC, and in the case of chiral products, optical rotation. Absolute stereochemistry of chiral products was determined by Mosher ester analysis[35]. Stock solutions of amino acid analogs were prepared in Milli-Q water at 200 mM concentrations. Metabolic labeling reporter β-ethynylserine (βES) is available from the authors on request. For detailed synthetic procedures, see Supplementary Note 1 in the Supplementary Information.

### Cell culture media

Dulbecco's modified Eagle's medium (DMEM, Thermo Fisher, Cat. No. 41965062) was used as complete medium used for metabolic labeling experiments with βES or HPG in HeLa cells (ATCC), while SILAC RPMI 1640 (RPMI 1640, cat. No: A2494201, supplemented with 300 mg/L L-Glutamine, 200 mg/L L-Arginine, 2.00 g/L D-Glucose, was used for metabolic labeling in Ramos B cells (obtained from Professor R.E.M. Toes, Leiden University Medical Center, Netherlands). Media were supplemented with 10% FCS and 100 units/mL penicillin and 100 μg/mL streptomycin before use. Methionine-free medium and threonine-free medium were custom-made in-house. Their formulations were based on DMEM and are outlined in Supplementary Table 2. Methionine-free and threonine-free media were sterile filtered and supplemented with 10% dialyzed FBS (Fisher Scientific, art. No. 15605639) and 100 units/mL penicillin and 100 μg/mL streptomycin before use.

### Cell lines and culture conditions

HeLa cells were cultured in Dulbecco's modified Eagle's medium (DMEM, Cat No. 41965062) supplemented with 10% fetal calf serum (FCS), 100 units/mL penicillin, and 100 μg/mL streptomycin. Ramos 3F3 cells[36] were cultured in RPMI 1640 medium (Fisher Scientific, Cat. No.: 11544526 supplemented with 10% FCS, 100 units/mL penicillin, and 100 μg/mL streptomycin. The cells were grown at 37 °C in humidified atmosphere of 5% $CO_2$ and were used within 15 passages for experiments.

### Amino acid analog labeling in B834 and BL21 E. Coli

10 mL of Lysogeny broth (LB) was inoculated from a glycerol stock of E. Coli B834 or BL21 and grown overnight in an orbital shaker at 37 °C. The cultures were diluted 10× with LB medium and the bacteria were grown at 37 °C until $OD_{600}$ was 0.6–0.8. Then, bacteria were collected by centrifugation (5000 × g, 3 min). E. coli B834 (a gift from S. van Kasteren, Leiden University) were incubated in SelenoMet medium (Molecular Dimensions) at 37 °C for 30 min to deplete intracellular methionine stores. Then, the bacteria were collected by centrifugation (5000 × g, 3 min) and were incubated at 37 °C for 1 h in 1 mL of SelenoMet medium supplemented with either 4 mM HPG or 4 mM βES and 4 mM methionine. E. coli BL21 were not starved and incubated directly at 37 °C for 1 h in 1 mL of LB supplemented with either 4 mM βES or 4 mM HPG. E. coli BL21 negative controls were incubated with 1 mL of either 4 mM βES and 34 μg/mL chloramphenicol (CAP) in LB medium or 4 mM βES and 200 mM threonine in LB medium. To assess βES incorporation over time, E. coli BL21 were incubated with 1 mL of 4 mM βES in LB medium for the indicated duration, after which βES incorporation was stopped by the addition of 34 μg/mL chloramphenicol. After incubation with amino acid analogs, both B834 and BL21 E. coli were collected by centrifugation, washed once with PBS, and resuspended in 100 μL of lysis buffer (50 mM HEPES, pH 7.4, 150 mM NaCl, 1% Triton-X-100, 1× EDTA-free protease inhibitor cocktail). Bacteria were lysed using a tip sonicator (MSE, Soniprep 150) for 3 × 10 s on ice. Lysate was

cleared by centrifugation (13,000 g, 30 min) at 4 °C, after which the supernatant was transferred to a new tube and the protein concentration was determined by BCA assay (Thermo Fisher). Lysates were flash-frozen and stored at −80 °C until further use.

### Bacterial growth assay

10 mL of LB was inoculated from a glycerol stock of E. Coli BL21 and grown overnight in an orbital shaker at 37 °C. The overnight culture was diluted 10× with LB medium and the bacteria were grown at 37 °C until $OD_{600}$ was 0.6–0.8. Then, bacteria were diluted to $OD_{600}$ 0.01, and 100 μL of this cell suspension was added to 100 μL of LB containing βES, chloramphenicol, or nothing (control). Final concentrations of βES and chloramphenicol were as indicated in the legend and caption of Fig. 2g. Cells were grown in a transparent 96-well plate at 37 °C and $OD_{600}$ was measured at indicated time points using a microplate reader (Biotek 800TS). Background absorbance present at $t = 0$ was subtracted from all data points.

### Metabolic labeling of HeLa cells for in-gel analysis

HeLa cells were seeded at a density of $3 \times 10^5$ cells per well in a six-well plate and grown for 1d in complete medium (DMEM). Then, for labeling experiments performed in threonine-free or methionine-free medium, HeLa cells were starved of intracellular threonine or methionine by incubating at 37 °C in threonine-free or methionine-free medium for 30 min. Then, cells were incubated at 37 °C for 1 h with 4 mM βES or 4 mM HPG in threonine-free or methionine-free medium, respectively. HeLa cells labeled in complete medium (DMEM) were not starved, but were directly incubated with 4 mM βES or 4 mM HPG in complete medium (DMEM) at 37 °C for 1 h. Following incubation, cells were washed with PBS, trypsinized, and washed with PBS again. Then, cells were resuspended in 100 μL of lysis buffer (50 mM HEPES, pH 7.4, 150 mM NaCl, 1% Triton-X-100, 1× EDTA-free protease inhibitor cocktail (Roche)) and lysed using a tip sonicator (MSE, Soniprep 150) for 3 × 10 s on ice. Lysate was cleared by centrifugation (13,000 × g, 30 min) at 4 °C, after which the supernatant was transferred to a new tube and the protein concentration was determined by BCA assay (Thermo Fisher). Lysates were flash-frozen and stored at −80 °C until further use.

### In-gel analysis of E. coli and HeLa lysates

100 μg of protein lysate was transferred to a new tube and was diluted to 90 μL total volume using lysis buffer (50 mM HEPES, pH 7.4, 150 mM NaCl, 1% Triton-X-100, 1 × EDTA-free protease inhibitor cocktail). Then 10 μL of a 10× concentrated CuAAC reagent mixture, consisting of 10 mM $CuSO_4 \cdot 5 H_2O$, 20 mM Tris-hydroxypropyltriazolylmethylamine (THPTA), 10 μM Sulfo-Cy5-azide (Jena Bioscience) and 100 mM sodium ascorbate, was added and the reaction mixture was agitated at 37 °C in an orbital shaker at 550 rpm for 30 min. Then, protein lysate was precipitated by the addition of 400 μL of ice-cold acetone and protein pellet was collected by centrifugation (5000× g, 20 min) at 4 °C. Protein pellet was allowed to dry on air and was dissolved in 1× Laemmli sample buffer (Bio-rad). Samples were heated at 70 °C for 10 min and 20 μg of protein for each sample was loaded onto a 1.5 mm 12% polyacrylamide gel. Gels were run at 80 V for the stacking gel and 120 V for the running gel. All samples were subjected to identical CuAAC labeling conditions, including all negative controls. In-gel fluorescence was measured with a Typhoon™ 5 gel scanner (Cytiva) using the 635 nm laser and the 670BP30 filter. Total protein amount was visualized by silver stain and imaged on a GelDoc XRS+ (Bio-rad). Relative intensity bar charts below gels show relative intensity of fluorescent signals normalized to corresponding silver stain signals. For normalization, fluorescent and silver stain intensity of full lanes was quantified using GelQuant.NET software (biochemlabsolutions.com). Uncropped versions of the gel images shown throughout the manuscript are provided in source data.

## Metabolic labeling of HeLa cells for flow cytometry

HeLa cells were seeded at a density of 60,000 cells per well in a 48-well plate and grown for 1 d in complete medium (DMEM). Then, for labeling experiments performed in threonine-free or methionine-free medium, HeLa cells were starved of intracellular threonine or methionine by incubating at 37 °C in threonine-free or methionine-free medium for 30 min. Then, cells were incubated at 37 °C for 1 h with various concentrations of βES or HPG in threonine-free or methionine-free medium, respectively. Negative controls for βES labeling were supplemented with either 100 μM cycloheximide or an excess (1 mM) of threonine. HeLa cells labeled in complete medium (DMEM) were not starved, but were directly incubated with various concentrations of βES or HPG at 37 °C for 1 h. After incubation with analog, all cells were chased by incubation in complete medium (DMEM) for 1 h at 37 °C. For time course analysis of incorporation, HeLa cells were incubated with 4 mM βES or 4 mM HPG for the indicated duration and cells were rinsed once with complete medium (DMEM) and labeling was stopped abruptly by addition of 600 μM cycloheximide. Following chase or cycloheximide treatment, cells were washed twice with cold PBS, trypsinized and fixed with 4% paraformaldehyde for 10 min. Cells were washed twice with PBS and stored at 4 °C until further use.

## Metabolic labeling of HeLa cells for microscopy

An eight-well chamber slide (Ibidi) was coated by treatment with 0.1% gelatin for 30 min at room temperature. The gelatin solution was removed and HeLa cells were seeded at a density of 50,000 cells per well and grown in complete medium (DMEM) at 37 °C for 1 d. Then, cells were incubated for 10 min in complete medium (DMEM) supplemented with either 4 mM βES or 4 mM HPG. Control cells were left untreated. Amino acid labeling was stopped by quickly aspirating the medium and replacing it by complete medium (DMEM) containing 600 μM cycloheximide. Then, cells were washed twice with cold PBS and fixed with 4% paraformaldehyde for 10 min. Cells were washed twice with PBS and stored at 4 °C until further use.

## Competition assay of analogs versus natural amino acids

HeLa cells were seeded at a density of 60,000 cells per well in a 48-well plate and grown for 1 d in complete medium (DMEM). Cells were starved of intracellular threonine or methionine by incubating at 37 °C in threonine-free or methionine-free medium for 1 h. Then, cells were incubated at 37 °C for 1 h with 50 μM βES and various ratios of threonine or 50 μM HPG and various ratios of methionine in threonine-free or methionine-free medium, respectively. After incubation, cells were washed twice with cold PBS, fixed with 4% paraformaldehyde for 10 min, washed twice with PBS, and stored at 4 °C until further use. Incorporated analogs were conjugated to Cy5-azide and quantified by flow cytometry.

## Cy5 conjugation to HeLa proteome via azide-alkyne click chemistry

Metabolic labeling for flow cytometry and microscopy was performed as previously described. Then, cells were permeabilized by incubation in 200 μL of 0.3% saponin in PBS for 10 min. Next, cells were incubated in 200 μL of a blocking buffer (0.1% saponin, 3% BSA in PBS) for 30 min. Copper-catalyzed azide-alkyne click (CuAAC) reaction was performed for 30 min at 37 °C with 100 μL of a CuAAC reaction mixture (4 mM CuSO4 · 5 H$_2$O, 200 μM THPTA, 0.5–5 μM sulfo-Cy5-azide and 40 mM sodium ascorbate, 0.1% saponin in PBS). For the experiments outlined in Fig. 3a, c–e, Supplementary Figs. 3–5 and Supplementary Fig. 8, 0.5 μM sulfo-Cy5 was used, while 5 μM sulfo-Cy5 was used for the experiment outline in Fig. 3b. Then, cells were washed 4× with 200 μL blocking buffer (0.1% saponin, 3% BSA in PBS) and resuspended in 3% BSA in PBS for flow cytometry or resuspended in PBS for widefield microscopy. All replicates were subjected to identical CuAAC labeling conditions, including all negative controls.

## Cell viability assay

HeLa cells were seeded at a density of 10,000 cells per well into a 96-well plate and grown for 1 d in complete medium (DMEM). Cells were then incubated for the indicated amount of time with various concentrations of βES in complete medium (DMEM) or various concentrations of HPG in methionine-free medium. Cells grown in complete medium (DMEM) without analog were used as a control. Following incubation, cells were trypsinized and resuspended in BD FACS flow buffer supplemented with 3 μM propidium iodide (Sigma). Propidium iodide exclusion was analyzed by flow cytometry.

## CellTrace violet assay

$1 \times 10^6$ HeLa cells were suspended in 10 mL of a 1 μM CellTrace violet solution in PBS and incubated at 37 °C for 20 min. Excess CellTrace dye was quenched by addition of 10 mL of complete medium (DMEM). Cells were split and resuspended in either complete medium (DMEM) without analog, complete medium (DMEM) supplemented with various concentrations of βES, or methionine-free medium supplemented with various concentrations of HPG. Cells were seeded at a density of $1 \times 10^5$ cells per well in a 12-well plate and incubated at 37 °C for 24 h. After trypsinization, cell proliferation was assessed by quantifying residual mean CellTrace fluorescence by flow cytometry. CellTrace-labeled HeLa cells at $t = 0$ were measured to establish baseline fluorescence.

## Flow cytometry

Flow cytometry was performed on a FACSVerse™ (BD Biosciences) flow cytometer. Cy5 was detected using the 633 nm laser line and the 660/10 filter, CellTrace violet using the 405 nm laser line and the 448/45 filter, and propidium iodide using the 488 nm laser and the 700/54 filter. Where deemed appropriate, flow cytometry laser voltage was lowered from default settings to avoid detector oversaturation. All samples within single experiments were measured using identical laser settings. The general gating strategy employed for all experiments is exemplified in Supplementary Fig. 23.

## Widefield microscopy

HeLa cells were metabolically labeled and subjected to CuAAC with Cy5-azide as previously described. Cells were then incubated with 300 nM 4′,6-diamidino-2-phenylindole (DAPI) solution in PBS for 3 min in the dark. Cells were washed twice with PBS and imaged on a DMi8 widefield microscope (Leica). Cy5 was detected using the 640 nm LED and the 666–724 nm filter, DAPI was detected using the 395 nm LED and the 420–450 nm filter. Images were processed and analyzed in ImageJ/Fiji (National Institutes of Health). Image contrast was enhanced to improve signal visibility by changing the minimum and maximum displayed values. Identical image contrast settings were used for images displayed within the same figure panel.

## BES Labeling of HeLa cells for determination of relative incorporation rate

HeLa cells were seeded at a density of $1 \times 10^6$ cells per T-75 flask and grown at 37 °C for 1 d. Cells were washed with PBS and incubated in threonine-free (DMEM) medium supplemented with 1 mM βES and 1 mM $^{13}C_4,^{15}N$-threonine (Thr$_5$, Cambridge Isotope Laboratories, CNLM-587-PK) at 37 °C for 72 h. Cells were washed with PBS, trypsinized, and collected by centrifugation. After washing the cells again in PBS, the PBS was aspirated and cells were flash-frozen and stored at −80 °C until further use. The experiment was performed with three replicates per condition.

## BES Labeling of HeLa cells for determination of proteomic response to βES

HeLa cells were seeded at a density of $1 \times 10^6$ cells per T-75 flask and grown at 37 °C for 1 d. Cells were washed with PBS and incubated in

complete medium (DMEM) with or without 1 mM βES at 37 °C for 5 h. Cells were washed with PBS, trypsinized, and collected by centrifugation. After washing the cells again in PBS, the PBS was aspirated and cells were flash-frozen and stored at −80 °C until further use. The experiment was performed once, without biological replicates.

## THRONCAT and BONCAT in HeLa cells for proteomics

HeLa cells were seeded at a density of $4 \times 10^6$ cells per T-75 flask and grown at 37 °C for 1 d. For HPG labeling, cells were incubated in methionine-free medium at 37 °C for 30 min to deplete intracellular methionine. Then, cells were incubated for 5 h at 37 °C with 4 mM HPG in methionine-free medium. HeLa cells labeled with βES were not starved, but instead incubated directly in either 1 mM βES or 4 mM βES in complete medium (DMEM) for 5 h at 37 °C. Untreated (control) cells were incubated for 5 h at 37 °C in complete medium (DMEM) without analog. After incubation, cells were washed twice in PBS, trypsinized and collected by centrifugation. After washing the cells again in PBS, the PBS was aspirated and cells were flash-frozen and stored at −80 °C until further use. The experiment was performed with three replicates per condition.

## THRONCAT labeling in Ramos cells for proteomics

$1 \times 10^7$ Ramos 3F3 cells were collected for each replicate by centrifugation. For steady-state replicates, cells were resuspended in 5 mL of SILAC RPMI 1640 (RPMI 1640, cat. No: A2494201, supplemented with 300 mg/L L-Glutamine, 200 mg/L L-Arginine, 2.00 g/L D-Glucose and 10% dialyzed FBS) with 146 mg/L $^{13}C_6,^{15}N_2$-L-lysine (Lys$_8$, Silantes, Prod. No 211604102) and 1 mM βES and grown at 37 °C for 1 h. Simultaneously, for anti-IgG stimulated replicates, cells were resuspended in RPMI 1640 supplemented with 10 μg/mL of anti-IgG (Invitrogen, Cat. No. 31143, 1/232 dilution). For anti-IgG stimulated replicates that were pulse-labeled between 1–2 h or 3–4 h, cells were resuspended in RPMI 1640 with 10 μg/mL anti-IgG and grown at 37 °C for 1 or 3 h, followed by centrifugation and incubation at 37 °C in SILAC RPMI 1640 with 10 μg/mL anti-IgG, 146 mg/L Lys$_8$ and 1 mM βES. For anti-IgG stimulated replicates that were pulse-labeled between 0–1 h, cells were resuspended in SILAC RPMI 1640 with 10 μg/mL anti-IgG, 146 mg/L Lys$_8$ and 1 mM βES and grown at 37 °C for 1 h. For all replicates, pulse labeling was stopped by the addition of 600 μM cycloheximide and cells were collected by centrifugation. Cells were washed once with PBS and cell pellets were flash frozen and stored at −80 °C until further use.

## Cell lysis for proteomics analysis

Cells were metabolically labeled as previously described. Then, cell pellets were resuspended in SDS lysis buffer (4% SDS, 1 mM DTT, 100 mM Tris-HCl pH 7.5) and were incubated for 5 min at 95 °C. Samples were sonicated until homogeneous using alternating cycles of 30 sec on/30 sec off on the highest setting using a BioRuptor Pico (Diagenode) and spun down at $16,000 \times g$ for 5 min. The supernatant was transferred to a new tube and protein concentrations were determined using Pierce BCA protein assay (ThermoFisher, 23225). Samples were stored at −80 °C until further use.

## Sample preparation for LC-MS/MS analysis of βES-/Thr$_5$-labeled HeLa lysates

Whole-cell lysates were prepared as previously described. For determination of βES incorporation rate relative to Thr$_5$, per sample 50 μg of total protein was taken for acetone precipitation by adding 10 volumes of ice-cold acetone and incubating for 10 min at −20 °C, followed by centrifugation for 10 min at max. speed. The protein pellet was washed with 100 μL ice-cold acetone and centrifuged for 10 min at max. speed. The resulting pellet was dissolved in 100 μL of 8 M urea, pH 8.0 at room temperature and the DTT concentration was adjusted to 10 mM. After 10 min incubation at room temperature, iodoacetamide (IAA) was added to a final concentration of 50 mM, followed by 10 min

incubation at room temperature in the dark. Per sample, 0.25 μg LysC was added and samples were incubated for 2 h at room temperature, after which three volumes of 50 mM ammonium bicarbonate and 0.1 μg trypsin were added, followed by overnight digestion. Next, samples were acidified with 10–20 μL trifluoroacetic acid (TFA) and concentrated on stage tips[37]. For determination of the proteomic response to βES labeling, 30 μg protein per sample was digested with mass spectrometry grade trypsin (Promega) using filter-aided sample preparation (FASP)[38], followed by desalting and concentration on C18 StageTips without acidification[37]. Peptides were then subjected to dimethyl labeling[39] and StageTips were stored at 4 °C until measurement by LC-MS/MS.

## Enrichment of newly synthesized proteins and on-bead digestion

Whole-cell lysates were prepared as previously described. Then, 400 μg and 200 μg of whole cell lysate in 250 μL PBS were used as input material for HeLa cells and Ramos cells, respectively. CuAAC reaction was performed using the materials and protocol from the Click-iT Protein Enrichment Kit (Invitrogen, C10416), with the adjustment that azide-functionalized agarose beads (Sigma Aldrich, 900957) were used to enrich the alkyne-labeled newly synthesized proteins, instead of the alkyne-functionalized agarose beads supplied with the Click-iT Protein Enrichment Kit. 100 μL bead slurry was used per reaction, which was washed with 1 mL H$_2$O, after which the cell lysate, 250 μL urea buffer, 500 μL 2× catalyst solution were added and incubated for 16–20 h at room temperature with end-over-end rotation, after which the beads were washed with 1 mL of H$_2$O. Then, beads were incubated with 10 mM DTT in 0.5 mL SDS buffer (provided by the kit) at 70 °C for 15 min while shaking. The beads were spun down, the supernatant was aspirated and the beads were incubated with 50 mM iodoacetamide in 0.5 mL SDS buffer for 30 min at room temperature while shaking. The beads were transferred to spin columns (provided by the kit) and washed with 20 mL of SDS buffer, 20 mL of 8 M urea in 100 mM Tris, pH 8, 20 mL of 20% isopropanol, 20 mL of 20% acetonitrile and 5 mL of PBS. The bound proteins were digested by resuspending the beads in 200 μL freshly prepared digestion buffer (2 M Urea, 100 mM Tris-HCl pH 8, 100 mM DTT) with 0.5 ug trypsin (Promega, V5111) and overnight incubation at room temperature while shaking. The digest was collected and acidified with 10 μL 10% TFA, after which the peptides were desalted and stored on StageTips[37].

## LC-MS/MS measurements and data analysis

Peptide samples were eluted from StageTips with elution buffer (80% acetonitrile, 0.1% formic acid in H$_2$O). For determination of βES incorporation rate relative to Thr$_5$, peptides were reduced to 10% of the original volume by vacuum concentration, diluted in 0.1% formic acid, and separated using an Easy-nLC 1000 liquid chromatography system (ThermoScientific) with a 94 min acetonitrile gradient (9–32%), followed by washes at 50 and 95% acetonitrile for a total of 120 min data collection. For analysis of enriched newly synthesized proteins from HeLa and Ramos cells, peptides were reduced to 10% of the original volume by vacuum concentration, diluted in 0.1% formic acid, and separated using an Easy-nLC 1000 liquid chromatography system (ThermoScientific) with a 44 min acetonitrile gradient (7–30%), followed by washes at 60 and 95% acetonitrile for a total of 60 min data collection. For determination of the proteomic response to βES labeling, peptide samples were eluted from StageTips with elution buffer (80% acetonitrile, 0.1% formic acid in H$_2$O). Then, light and medium-labeled samples for the forward and reverse reactions were combined. Next, samples were reduced to 10% of the original volume by vacuum concentration and diluted in 0.1% formic acid. Sample (5 μL) was injected and peptides were separated using an Easy-nLC 1000 liquid chromatography system (ThermoScientific) with a 44 min acetonitrile gradient (7–30%), followed by washes at 60 and 95%

acetonitrile for a total of 60 min data collection. In all cases, peptides were separated on an LC fitted with a 30 cm objective emitter of fused silica with an inner diameter of 75 µm packed with C18 beads (ReproSil-Pur, 1.9 µm, 120 Å) from Dr. Maish at a flow-rate of 250 nL/min.

Data-dependent measurements of the peptides were performed on an Orbitrap Exploris 480 (ThermoScientific). MS1 mass resolution was set to 120,000, the MS1 scan range was 350–1300 m/z, and MS/MS scan resolution was 15,000. Collision-induced dissociation energy was set at (N)CE 28. Automatic gain control (AGC) was set at $3.00 \times 10^6$ and $7.5 \times 10^4$ for MS1 and MS/MS, respectively. The AGC intensity threshold for MS/MS was set at $5.00 \times 10^4$. Precursors with charge states of 2–5 were selected for fragmentation. For every full scan, the top 20 peptides were selected for fragmentation, and dynamic exclusion was set to 45 seconds. Protein identification and quantification were done in MaxQuant v1.6.0.1[40] with match-between-runs, iBAQ, and label-free quantification enabled. In all cases, carbamidomethylation was specified as a fixed cysteine modification, and N-terminal acetylation and methionine oxidation were set as variable modifications. The MS/MS spectra were searched against the human Uniprot database including reverse peptide sequences for FDR estimation downloaded in June 2017. Digestion by trypsin/P was selected with the minimum peptide length set at 7, and up to 2 missed cleavages were allowed. Mass tolerance was set at 4.5 ppm and 20 ppm for precursor ion and fragment ions, respectively. FDR was set at 0.01 for both the peptide and protein levels. A minimum of two ratio counts were required for protein quantification.

For the determination of βES incorporation rate relative to $Thr_5$, Thr-to-βES (+9.984 Da) and Thr-to-$Thr_5$ (+5.0145 Da) were added to the default variable modifications. The MaxQuant 'modificationSpecificPeptides' output file was used to identify peptides that are present in both the $Thr_5$ and βES form to establish matched intensity ratios. To this end, all tryptic peptides containing one threonine residue that were detected in at least two replicates were selected and separated in $Thr_5$, βES, and $Thr_0$ groups. Next, peptide intensities were summed if a peptide was present in multiple modification states (e.g., both βES/Acetyl and βES) to obtain a single intensity value for each peptide. The 124 peptides that were detected in all three groups were selected for further analysis. Imputation for missing values was performed using the 'MinProb' method from the R DEP package (version 1.14.0)[41]. The peptide intensity values of individual replicates were then averaged per group and pairwise intensity ratios between βES- and $Thr_5$-peptides were calculated.

For determination of the proteomic response to βES labeling, carbamidomethylation was specified as fixed cysteine modification, and N-terminal acetylation, methionine oxidation, and threonine-to-βES (+9.984 Da) were set as variable modifications. Light (+0) and medium (+4) dimethyl labeling on the N-termini and lysine residues was specified under "labels". The MS/MS spectra were searched against a human UniProt database downloaded in June 2017. Common contaminants and decoy database hits were removed from the resulting MaxQuant output files and alias gene names were replaced with official gene symbols using the Limma package (version 3.48.3)[42]. If this resulted in duplicated entries, only the entry with the highest Andromeda Score was retained. Statistical significance thresholds were set at $Q3 + 1.5 \times IQR$ and $Q1 - 1.5 \times IQR$, and proteins were required to exceed these thresholds in both the forward and reverse run to be labeled as significant.

For analysis of enriched newly synthesized proteins from HeLa and Ramos cells, methionine-to-HPG (−22.0702 Da; HeLa experiment) and threonine-to-βES (+9.984 Da; HeLa and Ramos experiment) were added to the default variable modifications. For proteomic analysis of Ramos NSPs, heavy arginine ($Arg_6$) and $Lys_8$ were set as labels to exclusively use $Lys_8$-containing peptides for protein identification and quantification. The MS/MS spectra were searched against a human UniProt database downloaded in June 2017. Common contaminants

and decoy database hits were removed from the resulting MaxQuant output files and alias gene names were replaced with official gene symbols using the Limma package[42]. If this resulted in duplicated entries, only the entry with the highest Andromeda Score was retained. Only proteins that were detected in all replicates of a condition were marked as identified protein. For HeLa proteomic analysis, all proteins that did not show a significant enrichment (FC > 1.5, $p$.adj < 0.05) in the experimental conditions (THRONCAT/BONCAT) over the untreated (control) condition were considered background binders and were removed for downstream analysis in all condition. Average threonine- or methionine content of protein fractions was determined by a Python script (Supplementary Software). Uniprot reference proteome (UP000005640) was used as complete human proteome to analyze average proteomic threonine and methionine content.

Overrepresentation analysis was performed with DAVID 2021[43]. All detected proteins that were identified by βES and HPG labeling in the HeLa experiments were used as input list, and tested for overrepresentation in the gene sets belonging to the Gene Ontology: Cellular Compartment category C5 collection.

Differentially enriched protein analysis for the Ramos experiment was performed using the DEP package[41]. All proteins that were detected in all replicates of at least one condition were considered for downstream analysis. Imputation of missing values was performed using the MinProb method with the default settings. Imputation was performed 1000× and adjusted $p$ values and fold changes in LFQ intensities were calculated for each round. All proteins that showed an adjusted $p$ value < 0.05 and a fold change >1.5 in >80% of the iterations were considered to be significantly differentially expressed.

The mass spectrometry proteomics data have been deposited to the ProteomeXchange Consortium (http://proteomecentral.proteomexchange.org) via the PRIDE partner repository[44] with the dataset identifier PXD032368.

## Drosophila genetics
Flies were housed in a temperature-controlled incubator with 12:12 h on/off light cycle at 25 °C. *OK371-GAL4*, UAS-mCD8::GFP; UAS-mCD8::GFP was kindly provided by Dr. M. Freeman (Vollum Institute, Oregon Health & Science University, USA). The UAS-hGlyRS_G240R line was previously described[23]. UAS-hGlyRS_G240R or w1118 (control) flies were crossed with *OK371-GAL4*, UAS-mCD8::GFP; UAS-mCD8::GFP flies and larval offspring were used in THRONCAT experiments. A mixture of males and females was used in every experiment, except for the evaluation of motor performance (Supplementary Fig. 21e), where we used exclusively males due to gender-based differences in motor behavior in *Drosophila melanogaster*.

## THRONCAT in Drosophila melanogaster
For in vivo THRONCAT, previously described FUNCAT procedures[22,23,25,45] were adapted. 8 h or 16 h (overnight) egg collections were performed and animals were raised on Jazz-Mix *Drosophila* medium consisting of brown sugar, corn meal, yeast, agar, benzoic acid, methyl paraben, and propionic acid (Fisher Scientific) at 25 °C. 72 h after egg laying (AEL), third instar larvae were transferred to βES or HPG-containing medium. The standard βES or HPG concentration used was 4 mM, except for the βES dosage-titration experiment (Fig. 5e). The standard exposure time to non-canonical amino acid was 48 h, except in the experiments shown in Fig. 5f, g, in which larvae were exposed to βES for different time frames (2 h, 4 h, 8 h, 16 h, or 48 h). Larval central nervous systems (CNSs) or larval body walls were dissected in ice-cold HL3 solution and fixed in 4% paraformaldehyde for 30 min at room temperature. After fixation, the tissues were washed 3 × 15 min with PBST (1× PBS pH 7.2 containing 0.2% Triton-X-100) and 3 × 15 min with PBS pH 7.2. Metabolically labeled NSPs were conjugated to tetramethylrhodamine 5-carboxamido-(6-azidohexanyl) (TAMRA-$N_3$, Invitrogen, T10182) via

CuAAC. The CuAAC reaction mix was assembled in a defined sequence of steps. THPTA ligand (200 µM), TAMRA-N$_3$ (2 µM), CuSO$_4$ solution (4 mM), and sodium ascorbate solution (40 mM) were added to PBS pH 7.2. After each addition the solution was mixed thoroughly for 10 sec and at the end for 30 sec using a high-speed vortex. Larval tissues were incubated with 500 µl of CuAAC reaction mix overnight at 4 °C on a rotating platform. The next day, the tissues were washed 3 × 15 min with PBS-Tween (1× PBS pH 7.2 containing 1% Tween-20) and 3 × 15 min with PBST. Finally, tissues were mounted in Vecta-Shield mounting medium (Biozol, VEC-H-1000-CE) and stored at 4 °C until imaging using a Leica SP8 laser scanning confocal microscope. For image acquisition, identical confocal settings were used for all samples of a given experiment. Fluorescence intensities of the TAMRA signal were quantified using ImageJ/FIJI software (National Institutes of Health). All cells of one motor neuron cluster (9–10 cells) per ventral nerve cord cell boundaries were manually outlined to measure mean intensity values. Averaged values of all cells within one motor neuron cluster from each ventral nerve cord were used as single data points for statistical analysis.

### In vivo cell-type-specific FUNCAT in *Drosophila melanogaster*

For FUNCAT analysis of protein synthesis[22,23,25,45], 8 h egg collections of *OK371-GAL4* > UAS-MetRS$^{L262G}$ embryos were performed and animals were raised on Jazz-Mix *Drosophila* medium (Fisher Scientific) at 25 °C. 72 h AEL, third instar larvae were transferred to 4 mM azidonorleucine (ANL)-containing medium for 48 h. Larval central nervous system (CNS) was dissected in ice-cold HL3 solution and fixed in 4% paraformaldehyde (PFA) for 30 min at RT. After fixation, the CNSs were washed 3 × 10 min with PBST (1× PBS pH 7.2 containing 0.2% Triton-X-100) and 3 × 10 min with PBS pH 7.8. Metabolically labeled proteins were tagged by 'click chemistry' (Copper-Catalyzed (3 + 2)-Azide-Alkyne-Cycloaddition Chemistry (CuAAC)) using the fluorescent tag TAMRA. The FUN-CAT reaction mix was assembled in a defined sequence of steps. TBTA (1:1000, final concentration 200 µm), TAMRA-alkyne tag (1:5000, final concentration 0.2 µm), TCEP solution (1:1000, final concentration 400 µm) and CuSO4 solution (1:1000, final concentration 200 µm) were added to PBS pH 7.8. After each addition the solution was mixed thoroughly for 10 sec and at the end for 30 sec using a high-speed vortex. Larval CNSs were incubated with 500 µl of FUNCAT reaction mix overnight at 4 °C on a rotating platform. The next day, the CNSs were washed 3 times with PBS-Tween and three times with PBST for 15 min. Finally, larval CNSs were mounted in VectaShield mounting medium (Biozol, VEC-H-1000-CE) and stored at 4 °C until imaging using a Leica SP8 laser scanning confocal microscope. Because the signal intensity of βES labeled motor neurons was much stronger than the signal intensity of ANL-labeled motor neurons, it was not possible to use identical confocal settings for image acquisition of βES (THRON-CAT) versus ANL (FUNCAT) labeled CNSs. For this reason, we quantified fluorescence intensities of motor neurons relative to the background fluorescence intensity for each sample using ImageJ/FIJI software (National Institutes of Health). The fold-increase of staining intensity in motor neurons over background was calculated for each sample, using the mean intensity of all cells within one motor neuron cluster from each ventral nerve cord as single data points. The fold-increase over background was plotted relative to the fold-increase of the 48 h βES treatment data, which were arbitrarily set at an average of 100%.

### *Drosophila* studies to evaluate βES toxicity

To evaluate potential toxicity of βES exposure in larvae, 8 h egg collections of wild-type embryos were performed and animals were raised on Jazz-Mix *Drosophila* medium (Fisher Scientific) at 25 °C. 72 h AEL, third instar larvae were transferred to medium containing 4 mM βES, 4 mM HPG, 4 mM ANL, or control medium for 48 h. At the end of this incubation period, larvae were immobilized in ice next to a ruler, and imaged using a Leica MC170 HD microscope camera using the same

settings for all larvae. The larval body length was determined using ImageJ/FIJI software (National Institutes of Health).

The surface area of motor neuron cell bodies was measured using ImageJ/FIJI software (National Institutes of Health) on images that were obtained in THRONCAT experiments. The same motor neuron clusters that had been analyzed by THRONCAT were used to determine the surface area of motor neuron cell bodies. The synapse length of the neuromuscular junction (NMJ) on larval muscle 8 and the surface area of muscle 8 was determined as described in the next paragraph. A separate cohort of larvae was monitored to evaluate (i) the occurrence of larval lethality during the 48 h exposure time to non-canonical amino acids, (ii) the time between the end of the non-canonical amino acid treatment and pupa formation, (iii) the time between pupa formation and adult eclosion.

To evaluate potential toxicity of βES exposure in adult flies, we exposed 1-day-old flies to 4 mM βES or control medium for 48 h, followed by incubation for 24 h on control medium and evaluation of motor performance as described below.

### Analysis of larval muscle surface area and NMJ morphology

To analyze the NMJ synapse length on larval muscle 8 and surface area of muscle 8, third instar larvae were dissected in HL3 buffer and fixed in Bouin's for 3 min. After permeabilization and blocking (10% goat serum), immunostaining was performed with anti-Brp (anti-nc82; Developmental Studies Hybridoma Bank (DSHB), clone name: nc82, 1/100) or anti-Discs large 1 (anti-dlg1; DSHB, clone name: 4F3, 1/200) in larvae that selectively express membrane-tethered GFP in motor neurons (*OK371-GAL4*, UAS-mCD8::GFP; UAS-mCD8::GFP). Images were taken of muscle 8 in abdominal segment 5 using a Leica SP8 laser scanning confocal microscope with 20× Plan-Apochromat objective (0.8 NA). Maximum intensity projections of z-stacks comprising the entire NMJ were used to measure the synapse length and the surface area of muscle 8.

### Automated negative geotaxis assay to analyze motor performance

To assay motor performance, male flies were collected and divided into groups of 10 individuals. 4-day-old male flies were evaluated in a rapid iterative negative geotaxis assay (RING-assay), which had been previously established in our lab[15]. The assay is based on the innate escape response of flies to climb up the wall of a vial after being tapped down to its bottom. Flies were transferred into test tubes without anesthesia and six iterative measurements of groups of 10 flies per genotype were video recorded with a Nikon D3100 DSLR camera. The resulting movies were converted into 8-bit grayscale TIF image sequences with 10 frames/s. Subsequently, image sequences from runs 3, 4, and 5 were analyzed using an MTrack3 plug-in that automatically imports images in ImageJ, subtracts backgrounds, and filters and binarizes images to allow tracking of flies. Average climbing speed (mm/s) of all tracked flies was determined, averaged per test tube, and compared between treatment groups.

### Statistics

For flow cytometry experiments, at least three replicates were measured for each condition ($n = 3$). A minimum number of $n = 10,000$ events were measured (before gating) for each replicate. For experiments with *Drosophila melanogaster*, no statistical methods were used to pre-determine sample sizes, but sample sizes are similar to those reported in previous publications[22,23,25]. Whenever possible, data collection and analysis were performed by investigators blinded to the genotype of the experimental animals. Animals or samples were assigned to the various experimental groups based on the concentration and time frame of exposure to βES, HPG, or ANL, or their genotype. For a given experiment, samples from the different experimental groups were processed in parallel and analyzed in random

order. Before analysis, a Robust regression and Outlier removal method (ROUT) was performed to detect all outliers. Normality of all data was analyzed by Shapiro–Wilk, Anderson-Darling, and Kolmogorov–Smirnov tests. Subsequent statistical tests were only performed if all assumptions were met. Unpaired $t$ test was used for comparisons of two groups of normally distributed data with homogeneous variance. Mann–Whitney test was used for comparison of two groups of not normally distributed data. Brown-Forsythe ANOVA with Dunnett's T3 multiple comparisons test was used for comparison of more than two groups of normally distributed data with non-homogeneous variance. For the comparison of more than two groups of not normally distributed data, the Kruskal–Wallis test with Dunn's multiple comparisons test was used. Statistical analysis was performed using GraphPad Prism v.9.2.0.

### Reporting summary

Further information on research design is available in the Nature Portfolio Reporting Summary linked to this article.

## Data availability

The mass spectrometry proteomics data have been deposited to the ProteomeXchange Consortium via the PRIDE partner repository[44] with the dataset identifier PXD032368. Other data generated in this study are provided in the Supplementary Information/Source Data file or are available from the authors upon request. Source data are provided with this paper.

## Code availability

The custom script used to determine threonine- or methionine content of (sub-)proteomes is provided as Supplementary Software.

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

## Acknowledgements

This work is part of a project that has received funding from the European Research Council (ERC) under the European Union's Horizon 2020 research and innovation program (grant agreement No. 802940) and the NWO gravitation program 'Institute for Chemical Immunology' (NWO-024.002.009). E.St. is supported by an ERC consolidator grant (ERC-2017-COG 770244), and funding from the Radala Foundation, 'Stichting ALS Nederland', AFM-Telethon, ARSLA, the 'Prinses Beatrix Spierfonds' (W.OR22-03), the Muscular Dystrophy Association (MDA 946876) and an NWO Open Competition ENW-M grant. The Vermeulen lab is part of the Oncode Institute, which is partly funded by the Dutch Cancer Society (KWF).

## Author contributions

B.J.I. and K.B. designed the project. B.J.I. performed the organic synthesis and THRONCAT experiments in bacteria and HeLa cells. B.J.I. and M.J.v.W. performed THRONCAT labeling on Ramos B cells. J.D. and M.V. performed nascent protein enrichment, proteomics experiments, and analysis of proteomics data. E.Sl., N.M., and E.St. performed Drosophila experiments and analyzed the data. B.J.I. and K.B. wrote the main body of the manuscript. E.Sl., N.M., and E.St. wrote the Drosophila results and Drosophila methods sections of the manuscript. J.D. and M.V. wrote the proteomic methods of the manuscript. B.J.I., E.Sl., N.M., J.D., and E.St. designed graphical contents. All authors provided critical advice during the experimental phase and writing of the manuscript. All authors have given permission for publishing the manuscript. N.M. and E.Sl. contributed equally to the manuscript. E.St. and M.V. contributed equally to the manuscript.

## Competing interests

The authors declare no competing interests.
