## [Peer Review File · Nature Communications]

REVIEWER COMMENTS

Reviewer #1 (Remarks to the Author):

In this study, Ignacio et al. introduced β -ethynylserine as a threonine analog to label the newly-synthesized proteins. The authors showed that the β -ES could be incorporated into protein sequences in different organisms. With its clickable feature, this UAA coupled to a biotin conjugate enables to enrich the labeled proteomes for a MS analysis. The β -ES provides an alternative for methionine analogs, could be of interest to the researchers in protein dynamics. However, the current results don't convince me that β -ES is an efficient threonine substitute on the proteome scale. More, the authors tried to show that THRONCAT was better than methionine analogue-based approaches for NSP analysis throughout the manuscript, which is actually a bit misleading. AHA or HPG approaches are not as widely successful as pulsed-SILAC approach in dynamic studies for a few reasons, such as cellular toxicity, low insertion efficiency of UAAs, extra click chemistry and its imperfect reaction efficiency. Theoretically, the clickable-UAA-approach is more suitable for minor sub-proteome detection, such as secretome in a medium containing serum or short-time pulsed labeling, in which an enrichment step may help. However, THRONCAT of whole cellular NSPs using a long-term (5 hr) pulsed labeling may not have advantages over the popular p-SILAC strategy, even it was shown to be better than HPG. My other comments are listed below.

Major:

1. The incorporation efficiency of β -ES in different organisms (e.coli, HeLa and fly) should be critically evaluated. The fluorescence measured in Fig 3 doesn't support the fluorescent β -ES exists on proteins, all cellular non-protein-derived β -ES will impair this analysis. A more reasonable method is to check the ratios between threonine and β -ES in the proteomic data. This doesn't have to be analyzed on the paired peptides with threonine/ β -ES substitution. An overall estimation using site numbers of threonine and β -ES could provide substantial information. I searched three raw files (20210512_Exploris1_SA_Hela_ β ES_4mM_A/B/C) of HeLa cells (4mM β -ES treatment for 5 hours) using MaxQuant. A total number of 41450 peptides were identified, in which 20577 peptides contained at least one threonine site, whereas only 12 peptides were identified with a β -ES. This result clearly indicates the β -ES loses the competition between Thr and β -ES most of the time. A rough estimation of its insertion efficiency would be as low as $12/20577=0.06\%$.

The low incorporation efficiency of β -ES will lead biased results on NSP analysis. First, the extremely low chance to insert such an amino acid will definitely result in many NSPs without labeling. The insertion success rate is also highly affected by protein length (threonine number) and protein stabilities. More, if β -ES has an identical incorporation efficiency at all threonine sites is unknown.

Therefore, based on the authors' MS data, only an extremely small fraction of threonine sites could be inserted by the β -ES, making this strategy unsuitable for large scale studies.

2. Line 136 and Fig 3C. Related to the question 1. Since NSPs are produced in ribosome (located in cytosol), why was the strongest fluorescence observed in the nucleoli? I doubt that all these NSPs were just freshly expressed and transported to nucleoli, as it is well known that nucleosome/nucleoli proteins are extremely long-lived. Therefore, this observation actually supports the most fluorescence of is not originated from proteins. Authors should perform GO analysis to see if nucleoli proteins are enriched in their proteomic data. How many proteins in the pull-down using β -ES as a bait are really NSPs containing β -ES should be assessed.

3. Fig 3b. It's very surprising that β -ES doesn't incorporate more efficiently in Thr-free medium. Proteomic analysis should be done on cells cultured in threonine-free and complete media to see if the β -ES could be incorporated differentially between these two conditions.

4. Are there any side reactions could occur between β -ES and other biomolecules, such as the side chains of proteins?

5. Line 198 'For this, we stimulated Ramos B-cells with anti-IgG and pulse-labelled cells for 1 h at different time points with 1 mM β ES as well as d8-lysine, which provided a unique internal marker for NSPs.' More details on how the d8-lys data were analyzed and interpreted should be provided. First, the choose of d8-lys (deuterium) sounds uncommon. A more popular form of heavy Lys used in proteomics is $^{13}\text{C}_6$ or $^{13}\text{C}_6$ - $^{15}\text{N}_2$. D8-lysine will result in a significant RT shift for labeled peptides. However, the authors noted it again as $^{13}\text{C}_6$ - $^{15}\text{N}_2$ in the figure legend (Line 229), which was really confusing. Why did the authors choose to use trypsin, not Lys-C, to digest proteins if only the Lys was labeled? The authors should describe and explain their experiments clearly and precisely.

More importantly, the pulsed heavy Lys experiment can serve as a critical control dataset to assess the THRONCAT. The authors should at least perform the following analyses and add them in the manuscripts. 1) Show numbers of heavy-Lys-, light-Lys-, light-arg-containing peptides. The ratio (heavy-lys pep/ total lys pep) should be used to evaluate the performance of THRONCAT. Moreover, intensity ratios between paired peptides (heavy-lys vs. light-lys) should be calculated. 2) Count β -ES/Thr site numbers in heavy-Lys and light-Lys peptides. Is β -ES higher in heavy-Lys group? Is β -ES identified in light-Lys group? If yes, why?

6. It seems the authors only used identification, but not quantification, in the data analysis. This will lead to the loss of much valuable information. Quantification needs to be done.

Line 563. "Only proteins that were detected in all replicates of a condition were marked as identified protein. For HeLa proteomic analysis, all proteins that were detected in replicates of the untreated negative control were considered a-specific binders and were removed for downstream analysis for all conditions." This should be described more clearly. To define a specific binder, was any

quantitative threshold (ratio between β -ES and control) considered? How were fly and e.coli data processed? Removal of all proteins identified in control group will lose true binders identified in both groups but with significant enrichments in the β -ES group. Requiring detection in all replicates will lose low-abundance proteins. A-specific is a weird word.

7. The THRONCAT results should be correlated to proteins' stabilities/half-lives, which have been published many times elsewhere. Does THRONCAT have preference on proteins with different half-lives?

Minor:

1. Threonine/methionine concentration in culture media and fly food should be provided and compared to the supplementing β -ES.
2. The authors should measure the cellular concentrations of free amino acids (threonine and β -ES) at different incubation time points, as they make the precursor pool used for protein biosynthesis.
3. Line 558, why heavy arginine was searched?
4. The authors should try to see if peptide-based enrichment protocol can improve the site detection.
5. Fig2 d-g. Why fluorescent images were fully displayed, but the silver images were cropped?
6. To test the cellular response to β -ES, an omics analysis, e.g. proteomics or transcriptomics, is highly recommended.
7. Line 210-215. The authors should make the statement clear that this is only newly-expressed, not the total expression level.
8. Fig 1 and 2 should be combined.
9. Count Thr number for individual detected proteins and the whole proteome. Show the number distribution.

Reviewer #2 (Remarks to the Author):

I am enthusiastic in recommending publication of this paper in Nature Communications. The authors introduce a promising new reagent for time-resolved analysis of protein synthesis in prokaryotic and eukaryotic cells, and demonstrate important advantages of their method in comparison with

currently-available alternatives. If the authors can persuade a supplier to offer β -ethynylserine for purchase, it may become the reagent of choice for metabolic labeling of proteins in a wide variety of systems. I note the authors' generous offer to make the reagent available upon request.

I would ask the authors to consider just a few minor revisions:

1. It is not quite accurate to say that labeling with methionine analogs requires methionine starvation. In many cases (not all), reduction in the concentration of methionine in the medium is required for useful levels of labeling, but starvation is not. The authors' own results support this statement; Figure 3e shows that good labeling with HPG is accomplished at a roughly 30:1 ratio of analog to Met. This result is consistent with those reported by Bagert et al (Mol. Cell. Proteomics 13, 1352 (2014)). I would ask the authors to modify the language they use to address this issue, and I believe it would be helpful to cite the Bagert paper. Investigators who wish to use the BONCAT method should not be encouraged to starve their samples for methionine.
2. I don't believe the authors provide a description of the protocol used to acquire the results shown in Figure 4b and c. I'm curious about the difference in the numbers of proteins identified in these experiments and those reported in Figure 4d – i. Is the difference due to the labeling time? To the fact that isotope-labeling was used in the latter experiments but not the former? To some other aspect of the isolation protocol?
3. In line 532, the authors refer to "the provided beads substituted for azide agarose beads." I don't understand what this means. Can the authors provide a clearer description of the beads used for protein isolation?

Reviewer #3 (Remarks to the Author):

This manuscript describes a non-canonical threonine, β -ethynylserine (bES), for proteomic analysis of newly synthesized proteins (NSP). It does have advantages over other non-canonical Met analogs, which are the most popular AA for NSP studies and referred to as BONCAT. I think bES or THRONCAT could be very useful to many laboratories by itself or in combination with BONCAT.

1. Your data does not support your conclusions that bES does not affect cell growth. Supp Fig2 shows an obvious difference in growth at 2 and 3 hours with 4mM. Unfortunately, this expt was N=2 so no statistics can be performed. The control growth is also behaving very differently (plateauing) compared to the bES treated cells having a linear growth pattern. Since this is central to the manuscript, a statistical graph in the main text is needed. Cell growth is also perturbed in culture after 24hr, but no growth deficits were measured when *Drosophila* were labeled for 48 hours.

2. The silver stains in Figure 2 clearly demonstrate that you are comparing different amounts of total protein. To be fair and transparent, a graph depicting normalized values should be displayed. This is especially problematic for Fig2e in the comparison of bES and HPG and makes the line graph of the fluorescence very misleading. Also, how many times were these expts performed in Fig2? Figure 3b has the same problem with the silver stain and the number of biological expts is not stated.

3. Supplementary Fig 5 is not described accurately. In the main text, it is described as showing an increase in Thr-depleted media compared to complete media, but it is a comparison of bES and HPG. If I compare fig3a with complete media with Supp Fig 5 with Thr depleted media, they look almost identical.

4. No biological replicates are stated for Fig3F and it is unclear what is defined as normal and aberrant growth. Statistics should be used to define growth as aberrant.

5. In the comparison between bES and HPG in Hela cells, why was the 5 hour timepoint chosen? A typical BONCAT expt in cultured cells is 30min-60min since the media conditions are not optimal. Plus, the manuscript touts the fast kinetics of bES. A one hour comparison would be more informative to the reader in regards to proving fast kinetics as shown with BONCAT studies. The next expt is a 1hour bES treatment and the protein IDs are much lower than an expected for a BONCAT expt. If HPG incorporates better than bES, it does not take away the usefulness of bES, because bES can be used in complete media.

6. Line 186 states THRONCAT does not enrich for Thr containing proteins, but that is exactly what it does. Confusing statement.

7. The incorporation of bES in *Drosophila* is very interesting but incomplete. How well was the entire organism labeled? Were there any negative effects on the development of this complex organism? It is stated this expt was chosen because the authors previously published in Nature Communications the incorporation of ANL in *Drosophila*. It was disappointing that a bES and ANL comparison was not performed. In the ANL paper, the negative effects of ANL on *Drosophila* development was thoroughly investigated. It was a very well-done comprehensive study! With the negative growth effect of bES after 24hours, I would expect it would disturb normal fly development after 48 hours. I expect the same rigor as their previous Nature Communications paper.

8. Since this is a new compound for mass spectrometry analysis, it needs to be verified that the bES containing peptides have the expected mass shift.

9. AHA does compete with Met and is often coupled with Met starvation or depletion, but it has been successfully used without Met depletion. Please correct the text.

Liu et al. Elife 2018 Feb 7;7:e33420

Evans et al EMBO J 2019 Jul 1;38(13):e101174

10. Why was bES chosen over 4-FT or B-HNV?

11. Can bES be phosphorylated? There are many well-known thr kinases(i.e. MAPK) that are essential for cellular physiology. Please discuss in the text how you envision this could affect the application of THRONCAT.

Reviewer 1:

... “More, the authors tried to show that THRONCAT was better than methionine analogue-based approaches for NSP analysis throughout the manuscript, which is actually a bit misleading. AHA or HPG approaches are not as widely successful as pulsed-SILAC approach in dynamic studies for a few reasons, such as cellular toxicity, low insertion efficiency of UAAs, extra click chemistry and its imperfect reaction efficiency. Theoretically, the clickable-UAA-approach is more suitable for minor sub-proteome detection, such as secretome in a medium containing serum or short-time pulsed labeling, in which an enrichment step may help. However, THRONCAT of whole cellular NSPs using a long-term (5 hr) pulsed labeling may not have advantages over the popular p-SILAC strategy, even it was shown to be better than HPG. My other comments are listed below.”

>> *We thank the reviewer for the considerations and remarks on our manuscript. We agree with the reviewer that if no protein enrichment is needed for proteomic experiments, pulsed SILAC or similar, is an excellent method of choice with all the given arguments. However, in some experimental setups, protein enrichment greatly improves proteomic coverage by removing interference from other proteins that are not of interest. Examples of such experiments are analysis of fast cellular responses to external stimuli (where ‘old’ cellular proteins are present), secreted proteins (where exogenous proteins are present in culture media and as also pointed out by the reviewer), or analysis of (intercellular) exchange of proteins (where other intracellular proteins are present). Besides these, THRONCAT has potential biotechnological applications where facile introduction of a bioorthogonal handle may be beneficial, which can now be achieved by simply adding β ES to the experimental cell culture media. We have included some additional sentences throughout the manuscript to specify possible applications of labeling proteomic subsets. To further address the issues raised by the reviewer we have performed and included additional experiments to show that β ES is efficiently incorporated into proteins as outlined below.*

...“ . The incorporation efficiency of β ES in different organisms (e.coli, HeLa and fly) should be critically evaluated. The fluorescence measured in Fig 3 doesn't support the fluorescent β ES exists on proteins, all cellular non-protein-derived β ES will impair this analysis.

>> *We agree with the reviewer that fluorescence may potentially arise from non-protein-derived β -ethynyl serine (β ES) present in the cell when measured by FACS or fluorescence microscopy. However, we are confident that β ES is incorporated into proteins for several reasons. 1) The fluorescence measured in Fig. 3 are from fixed cells. Any remaining unincorporated β ES or fluorophore is removed in thorough washing steps. 2) In control experiments, we performed labeling experiments in the presence of protein synthesis inhibitor cycloheximide (CHX). Treatment with CHX strongly diminishes the fluorescent signal obtained by β ES labeling, indicating that active protein synthesis is required for β ES labeling and that β ES is incorporated into newly synthesized proteins (Supplementary Fig. 4). We also repeated these experiments on HeLa cells in the presence of protein synthesis inhibitor CHX and analyzed by FACS and fluorescence microscopy and included additional figures (revised manuscript: Fig. 3b, Fig. 3c and Supplementary Fig. 3, Supplementary Fig. 5). We verified incorporation of β ES into proteins, as evidenced by strong fluorescent labeling of the proteome on SDS-PAGE gel following click reaction with Cy5.*

...”A more reasonable method is to check the ratios between threonine and β ES in the proteomic data. This doesn't have to be analyzed on the paired peptides with threonine/ β ES substitution. An overall estimation using site numbers of threonine and β ES could provide substantial information. I searched three raw files (20210512_Exploris1_SA_Hela_ β ES_4mM_A/B/C) of HeLa cells (4mM β ES treatment for 5 hours) using MaxQuant. A total number of 41450 peptides were identified, in which 20577 peptides contained at least one threonine site, whereas only 12 peptides were identified with a β ES. This result clearly indicates the β ES loses the competition between Thr and β ES most of the time. A rough estimation of its insertion efficiency would be as low as $12/20577=0.06\%$.”

>> *We appreciate the analysis performed by the reviewer on our data and have to politely disagree with the conclusions of the reviewer. The LC-MS/MS data in the raw files stems from labeled proteins that have been enriched on azide-modified beads as described in the methods section; alkyne-labeled proteins are covalently conjugated to the azide beads via their β ES-modifications and, after tryptic digestion, tryptic peptides are eluted from the beads. It is therefore expected that most β ES modifications remain covalently attached to the beads and would not appear in the dataset analyzed by the reviewer. As such, the dataset analyzed by the reviewer is not suitable for determining the incorporation rate of β ES in the mammalian proteome. Yet, in order to gain more insight on the incorporation efficiency of β ES, we performed an additional experiment where we label cells with heavy Threonine (Thr_5) as well as β ES in a 1:1 ratio for 72 hours. We did not perform an enrichment step in this case, but instead analyzed whole-cell lysate containing the β ES-labeled peptides directly to compare the relative incorporation of threonine and β ES by mass spectrometry. By comparing the peak intensity ratio of 124 paired peptides with either a Thr_5 - or a β ES-modification, we established in these peptides that Thr_5 was incorporated ~ 20 - $60x$ more than β ES (mean 40.8), indicating an incorporation efficiency of $\sim 2.5\%$ for β ES compared to threonine. We have included this data (revised manuscript: Supplementary Fig. 7) and added an additional paragraph in the manuscript. For comparison, it is estimated that methionine is incorporated ~ 500 times more efficient than HPG in *E. coli*, indicating a relative incorporation efficiency of only $\sim 0.2\%$ as described by Kiick, K. L., Weberskirch, R. & Tirrell, D. A. Identification of an expanded set of translationally active methionine analogues in *Escherichia coli*. *FEBS Lett.* 502, 25–30 (2001).*

...”The low incorporation efficiency of β ES will lead biased results on NSP analysis. First, the extremely low chance to insert such an amino acid will definitely result in many NSPs without labeling. The insertion success rate is also highly affected by protein length (threonine number) and protein stabilities. More, if β ES has an identical incorporation efficiency at all threonine sites is unknown. Therefore, based on the authors' MS data, only an extremely small fraction of threonine sites could be inserted by the β ES, making this strategy unsuitable for large scale studies.”

>> *As described above, we determined that the relative incorporation efficiency of β ES is $\sim 2.5\%$ (or 1 in 40 residues) compared to threonine, at equimolar concentrations. In our NSP-enrichment experiment in HeLa cells we used a 5-fold higher concentration of β ES (4 mM) compared to threonine (0.8 mM in complete DMEM medium, see Supplementary Table 2), presumably increasing the actual incorporation rate into proteins even further. Although we agree with the reviewer that it is not known whether β ES has an identical incorporation efficiency at all threonine sites, we believe that the data in Fig. 5b – showing enrichment of 3559 HeLa proteins after labeling with 4 mM β ES vs. enrichment of only 501 proteins in a control experiment without β ES labeling (untreated) – strongly suggests β ES is widely incorporated into*

thousands of proteins, facilitating their enrichment. We note that we enriched a similar number of proteins using 4 mM homopropargylglycine (HPG) in methionine depleted culture media, which is widely accepted as a metabolic label with broad incorporation throughout the proteome. We also observed that THRONCAT enrichment is not biased towards threonine-rich proteins, since the average threonine content in THRONCAT-enriched proteins in HeLa cells is identical to that of the human proteome. We performed an analysis to determine whether THRONCAT-enrichment was biased towards short- or long-lived proteins and found no excessive bias regarding protein stability (Supplementary Fig. 15). We included a paragraph in the manuscript on this reasoning.

...”2. Line 136 and Fig 3C. Related to the question 1. Since NSPs are produced in ribosome (located in cytosol), why was the strongest fluorescence observed in the nucleoli? I doubt that all these NSPs were just freshly expressed and transported to nucleoli, as it is well known that nucleosome/nucleoli proteins are extremely long-lived. Therefore, this observation actually supports the most fluorescence of is not originated from proteins. Authors should perform GO analysis to see if nucleoli proteins are enriched in their proteomic data.”

>> *We thank the reviewer for the suggestion to do GO analysis. We have performed a GO analysis of the proteins that we identified after enrichment of HeLa NSPs using THRONCAT and also compared that to the proteins identified using BONCAT. In both cases nucleolar proteins were detected and THRONCAT- and BONCAT-enriched proteins show similar subcellular distributions. We have included this additional information in the manuscript (revised manuscript: Supplementary Fig. 13). In contrary to what the reviewer suggests, it is well-established that nucleoli are the site of rapid accumulation of ribosomal proteins, which are highly expressed (Lam, Y. W., Lamond, A. I., Mann, M. & Andersen, J. S. Analysis of Nucleolar Protein Dynamics Reveals the Nuclear Degradation of Ribosomal Proteins. Curr. Biol. 17, 749–760 (2007)). In addition, it is established that nucleoli have very high protein density, possibly explaining the high fluorescence intensity we and others have observed in fluorescence microscopy (Lam, Y. W., Lamond, A. I., Mann, M. & Andersen, J. S. Analysis of Nucleolar Protein Dynamics Reveals the Nuclear Degradation of Ribosomal Proteins. Curr. Biol. 17, 749–760 (2007)). Similar enhanced fluorescence intensity in nucleoli was previously observed using the BONCAT method and OPP labeling, which are established methods to label newly synthesized proteins (Beatty, K. E. et al. Fluorescence Visualization of Newly Synthesized Proteins in Mammalian Cells. Angew. Chem. Int. Ed. 45, 7364–7367 (2006) and Liu, J., Xu, Y., Stoleru, D. & Salic, A. Imaging protein synthesis in cells and tissues with an alkyne analog of puromycin. Proc. Natl. Acad. Sci. 109, 413–418 (2012)).*

..”How many proteins in the pull-down using β ES as a bait are really NSPs containing β ES should be assessed.”

>> *We refer to the experiments above where we compare the incorporation of β ES and heavy threonine in HeLa NSPs.*

...”3. Fig 3b. It’s very surprising that β ES doesn’t incorporate more efficiently in Thr-free medium. Proteomic analysis should be done on cells cultured in threonine-free and complete media to see if the β ES could be incorporated differentially between these two conditions.”

>> We agree with the reviewer that this is counterintuitive. We assign this observation to the high amount of β ES incorporated into the NSPs, combined with the limiting amount of fluorophore used in where we are limited to avoid high levels of background fluorescence due to potential dye aggregation and background binding to the samples. We have added a sentence to the manuscript to address this observation. We have repeated the same experiment, only now using FACS as a readout to more accurately quantify the incorporation of the analogs (revised manuscript: Fig. 3b).

..."4. Are there any side reactions could occur between β ES and other biomolecules, such as the side chains of proteins?"

>> The alkyne reactive handle present in β ES is an often used motif in bioorthogonal chemistry. The handle is biologically inert and does not react with any of the functionalities present in the cell and other biomolecules.

..."5. Line 198 'For this, we stimulated Ramos B-cells with anti-IgG and pulse-labelled cells for 1 h at different time points with 1 mM β ES as well as d8-lysine, which provided a unique internal marker for NSPs.' More details on how the d8-lys data were analyzed and interpreted should be provided. First, the choose of d8-lys (deuterium) sounds uncommon. A more popular form of heavy Lys used in proteomics is 13C6 or 13C6-15N2. D8-lysine will result in a significant RT shift for labeled peptides. However, the authors noted it again as 13C6-15N2 in the figure legend (Line 229), which was really confusing."

>> We thank the reviewer for noticing this mistake and sincerely apologize for the confusion. Indeed, we performed our experiments using 13C6-15N2 (e.g. Lys₈) instead of d8-lys (which indeed is uncommon). We have corrected the mistake in the manuscript and figures.

..."Why did the authors choose to use trypsin, not Lys-C, to digest proteins if only the Lys was labeled? The authors should describe and explain their experiments clearly and precisely."

>> Trypsin was used, as the combination of LC and MS that we use is optimized for peptides with lengths that result from trypsin digestion. The reviewer is correct that Lys-c would be better, if the mass spectrometry setup was optimized for this type of experiment. For our manuscript we concluded that trypsin is sufficient to obtain the results needed to support the statements made.

..."More importantly, the pulsed heavy Lys experiment can serve as a critical control dataset to assess the THRONCAT. The authors should at least perform the following analyses and add them in the manuscripts. 1) Show numbers of heavy-Lys-, light-Lys-, light-arg-containing peptides. The ratio (heavy-lys pep/ total lys pep) should be used to evaluate the performance of THRONCAT. Moreover, intensity ratios between paired peptides (heavy-lys vs. light-lys) should be calculated. 2) Count β ES/Thr site numbers in heavy-Lys and light-Lys peptides. Is β ES higher in heavy-Lys group? Is β ES identified in light-Lys group? If yes, why?"

>> We agree with the reviewer that this would give much information on the incorporation of β ES in the peptides. To address this issue, we have determined the number of heavy-lys (Lys₈) and light-lys (Lys₀) containing peptides in the Ramos experiment. As shown in the novel Supplementary Fig. 16, Lys₈-containing peptides are strongly overrepresented in THRONCAT-enriched proteins from Ramos cells, indicating THRONCAT enriched the newly synthesized proteins with high specificity.

6. It seems the authors only used identification, but not quantification, in the data analysis. This will lead to the loss of much valuable information. Quantification needs to be done. Line 563. “Only proteins that were detected in all replicates of a condition were marked as identified protein. For HeLa proteomic analysis, all proteins that were detected in replicates of the untreated negative control were considered α -specific binders and were removed for downstream analysis for all conditions.” This should be described more clearly. To define a specific binder, was any quantitative threshold (ratio between β ES and control) considered? How were fly and e.coli data processed? Removal of all proteins identified in control group will lose true binders identified in both groups but with significant enrichments in the β ES group. Requiring detection in all replicates will lose low-abundance proteins. A-specific is a weird word.

>> *As the key message in our manuscript is to present THRONCAT as novel tool that can be used to profile (changes in) the nascent proteome, we mostly presented our data in a qualitative manner throughout the main manuscript. However, we also quantified the data as shown in Fig. 4e-l and Supplementary Figs. 7, 10, 11 and 16 in the revised manuscript. For example, to determine specific binders, we now have performed quantitative analysis between the control and experimental labeling conditions as described in the experimental section, showing that none of the proteins detected in the control condition was significantly enriched in the β ES conditions, and that only two proteins were significantly enriched in the HPG condition (revised manuscript: Supplementary Fig. 11). $FC > 1.5$ and $p_{adj} < 0.05$ were used as significance cutoffs. Similarly, we performed quantitative analysis of B cell stimulation at different time windows and after imputation of the data as described in the experimental sections and shown in supplementary data 5 and as demonstrated in the main text Figure 4e-i. To demonstrate the sensitivity and to reduce the amount of data in the current manuscript, we chose to use very stringent selection conditions to confidentially identify a protein hit and significant changes therein. Analysis settings for future research efforts following this data can be adjusted accordingly and are beyond the scope of this manuscript.*

...”7. The THRONCAT results should be correlated to proteins’ stabilities/half-lives, which have been published many times elsewhere. Dose THRONCAT have preference on proteins with different half-lives?”

>> *We performed an analysis on protein turnover rates and compared the data to those reported by Zecha, J. et al. (Peptide Level Turnover Measurements Enable the Study of Proteoform Dynamics Mol. Cell. Proteomics 17, 974–992 (2018)) and Bagert, J. D. et al. (Quantitative, Time-Resolved Proteomic Analysis by Combining Bioorthogonal Noncanonical Amino Acid Tagging and Pulsed Stable Isotope Labeling by Amino Acids in Cell Culture. Mol. Cell. Proteomics 13, 1352–1358 (2014)). The detected proteins shows that THRONCAT has a similar profile as experiments done with AHA. Furthermore, proteins detected by HPG showed a similar profile as β ES proteins, further underscoring that THRONCAT has a comparable labeling profile as the current golden standard. We have included a section in the main manuscript and included an additional supporting figure (revised manuscript: Supplementary Fig. 14.*

Minor:

..."1. Threonine/methionine concentration in culture media and fly food should be provided and compared to the supplementing β ES."

>> *We have included the concentrations of threonine and methionine in the cell media in Supplementary Table 2. Concentrations of threonine and methionine in E. coli medium (LB) and Drosophila food mix are not provided by the supplier and therefore unknown.*

..."2. The authors should measure the cellular concentrations of free amino acids (threonine and β ES) at different incubation time points, as they make the precursor pool used for protein biosynthesis.

>> *We believe that measuring the intracellular concentration of free amino acids at different incubation time points would not be representative as the concentrations of free threonine and β ES amino acids would strongly vary on the type of cell or organism used, the culture medium, experimental setup and research question asked.*

..."3. Line 558, why heavy arginine was searched?"

>> *Heavy arginine was searched to exclude all arginine peptides from analysis. As only heavy lysine was used, we only used on lysine-peptides for identification and quantification. The reviewer is correct that the method could be even further improved if we included heavy arginine as well, but this would lead to lengthy optimization experiments. As this is a proof-of-principle study, we reasoned that only using heavy lysine would be sufficient.*

..."4. The authors should try to see if peptide-based enrichment protocol can improve the site detection."

>> *We agree with the reviewer that a peptide-based enrichment protocol, such as the DiDBit protocol described in Schiapparelli et al. (Schiapparelli et al. Direct detection of biotinylated proteins by mass spectrometry, 13, 3966-3978, (2014)) could increase detection of newly synthesized peptides. In our current experimental setup, a peptide-based enrichment protocol would not lead to any hits as we covalently ligate the peptides on beads. Using such protocol would require the use of cleavable linkers or modification with, for example, biotin-azide and pulldown with avidin beads. In theory this would be possible and interesting, but would require lengthy optimization protocols that is beyond the scope of this manuscript. Comparison of the different enrichment protocols using THRONCAT would be an interesting topic for a future manuscript.*

..."5. Fig2 d-g. Why fluorescent images were fully displayed, but the silver images were cropped?"

>> *We have used the silver staining purely as loading control and are therefore cropped. Full gels are included in the source data file.*

..."6. To test the cellular response to β ES, an omics analysis, e.g. proteomics or transcriptomics, is highly recommended."

>> *We have analyzed the proteomic response of HeLa cells to 1 mM β ES incubation for 5 h and found only 2 differentially expressed proteins compared to untreated HeLa cells. This data is presented in Supplementary Fig. 10 in the revised manuscript.*

7. Line 210-215. The authors should make the statement clear that this is only newly-expressed, not the total expression level.

>> *We thank the reviewer for the note and corrected the statement.*

8. Fig 1 and 2 should be combined.

>> *We kindly disagree with the reviewer as this would cover too much data in a single figure.*

9. Count Thr number for individual detected proteins and the whole proteome. Show the number distribution.

>> *We have analyzed the amount of threonines and the proteome using Python script as presented in Supplementary Data 3, we found that the THRONCAT-identified proteins had an average threonine content of 5.2%, which is similar to that the average threonine content of the complete mammalian proteome. We have added a sentence in our manuscript to address this point.*

Reviewer #2 (Remarks to the Author):

..."I am enthusiastic in recommending publication of this paper in Nature Communications. The authors introduce a promising new reagent for time-resolved analysis of protein synthesis in prokaryotic and eukaryotic cells, and demonstrate important advantages of their method in comparison with currently-available alternatives. If the authors can persuade a supplier to offer b-ethynylserine for purchase, it may become the reagent of choice for metabolic labeling of proteins in a wide variety of systems. I note the authors' generous offer to make the reagent available upon request.

1. It is not quite accurate to say that labeling with methionine analogs requires methionine starvation. In many cases (not all), reduction in the concentration of methionine in the medium is required for useful levels of labeling, but starvation is not. The authors' own results support this statement; Figure 3e shows that good labeling with HPG is accomplished at a roughly 30:1 ratio of analog to Met. This result is consistent with those reported by Bagert et al (Mol. Cell. Proteomics 13, 1352 (2014)). I would ask the authors to modify the language they use to address this issue, and I believe it would be helpful to cite the Bagert paper. Investigators who wish to use the BONCAT method should not be encouraged to starve their samples for methionine."

>> *We thank the reviewer for this comment and agree that complete starvation is not essential in all cases when using BONCAT. We see the THRONCAT method to be used complementary to BONCAT and have adjusted the text accordingly. We further included several examples where depletion is not required, including the paper of Bagert et al.*

..."2. I don't believe the authors provide a description of the protocol used to acquire the results shown in Figure 4b and c. I'm curious about the difference in the numbers of proteins identified in these experiments and those reported in Figure 4d – i. Is the difference due to the labeling time? To the fact

that isotope-labeling was used in the latter experiments but not the former? To some other aspect of the isolation protocol?"

>> *The followed steps are described under 'THRONCAT and BONCAT in HeLa cells for proteomics', 'Enrichment of nascent proteins and on-bead digestion', and 'LC-MS/MS measurement and data analysis' in the Methods & Materials section. The larger number of identified newly synthesized proteins (NSPs) in HeLa cells compared to Ramos cells can be attributed to both the longer labeling time (5 h vs. 1 h, resp.) and cell type (HeLa vs. B-cells, resp.) Moreover, Ramos NSPs were stringently filtered on a computational level by the requirement of having a heavy lysine (lys₈) label to remove potential background proteins, while HeLa NSPs were not. Furthermore, the proteins shown in fig. 4e only represent the Ramos NSPs with statistically significant differential expression upon B-cell stimulation, whereas Fig. 4b,c show all enriched HeLa NSPs.*

..."3. In line 532, the authors refer to "the provided beads substituted for azide agarose beads." I don't understand what this means. Can the authors provide a clearer description of the beads used for protein isolation?"

>> *We apologies that this part was not clearly described in the manuscript. The beads that come with the Click-iT (Invitrogen, C10416) kit are alkyne agarose beads that were designed for the enrichment of azide-labeled proteins in, for instance, BONCAT experiment. However, in our case, we used alkyne-labeled amino acids, requiring azide agarose beads for successful enrichment of the alkyne-labeled proteins. Therefore, we used alternative beads to supplement the Click-iT kit, keeping all other materials and steps of the enrichment protocol the same. For clarification, we have adjusted the text and provided detailed information on the availability of the beads in the methods section.*

Reviewer #3 (Remarks to the Author):

..."This manuscript describes a non-canonical threonine, β -ethynylserine (bES), for proteomic analysis of newly synthesized proteins(NSP). It does have advantages over other non-canonical Met analogs, which are the most popular AA for NSP studies and referred to as BONCAT. I think bES or THRONCAT could be very useful to many laboratories by itself or in combination with BONCAT.

1. Your data does not support your conclusions that bES does not affect cell growth. Supp Fig2 shows an obvious difference in growth at 2 and 3 hours with 4mM. Unfortunately, this expt was N=2 so no statistics can be performed. The control growth is also behaving very differently (plateauing) compared to the bES treated cells having a linear growth pattern. Since this is central to the manuscript, a statistical graph in the main text is needed. Cell growth is also perturbed in culture after 24hr, but no growth deficits were measured when Drosophila were labeled for 48 hours."

>> *We thank and agree with the reviewer on this point and repeated the work to validate that β ES does not affect cell growth in more detail and with more replicates and made the following observations: 1) we did not observe limited cell growth with E. coli using β ES. To support this conclusion, we included E. coli growth curves of 6 biological replicates (revised manuscript: Fig. 2g). 2) We further looked at proliferation of Hela cells for 24 hours using varying concentrations of β ES in triplicate (revised manuscript: Supplementary Fig. 6). Here, we observed a slight, but statistically significant, reduction in cell proliferation at the highest concentration used (4 mM β ES), while lower concentrations did not significantly affect cell proliferation. Significant cell proliferation was observed using HPG in depleted medium at all concentrations tested 3) we have novel data on the effect of 4 mM β ES incorporation on larval body length after 48 h (revised manuscript: Supplementary Fig. 20b), the cell body area of motor neurons in the ventral nerve cord (revised manuscript: Supplementary Fig. 20c), the length of the neuromuscular synapse on muscle 8 (revised manuscript: Supplementary Fig. 20d), and larval muscle 8 surface area (revised manuscript: Supplementary Fig. 20e). These novel data are discussed in detail in Supplementary Discussion 1. We found a tendency for reduced larval body length induced by β ES administration (which did not reach statistical significance), no effect on motor neuron cell body area, no effect on neuromuscular synapse length, and a 20% reduction of the surface area of larval muscle 8. Overall, these data indicate a limited reduction in larval growth induced by 48 h β ES incorporation.*

..."2. The silver stains in Figure 2 clearly demonstrate that you are comparing different amounts of total protein. To be fair and transparent, a graph depicting normalized values should be displayed. This is especially problematic for Fig2e in the comparison of bES and HPG and makes the line graph of the fluorescence very misleading."

>> *We have repeated the gels in Fig. 2 and make the loading more equal. We also included the normalized values on the band intensities in all figure panels with gels.*

..."Also, how many times were these expts performed in Fig2? Figure 3b has the same problem with the silver stain and the number of biological expts is not stated."

>> *The gels have been performed several times (at least n=2) with consistent outcomes. We have added the number of biological replicates to the respective figure captions. Representative gels are shown in the manuscript.*

..."3. Supplementary Fig 5 is not described accurately. In the main text, it is described as showing an increase in Thr-depleted media compared to complete media, but it is a comparison of bES and HPG. If I compare fig3a with complete media with Supp Fig 5 with Thr depleted media, they look almost identical."

>> *It must be noted that while Fig. 3a and Supplementary Fig. 5 look almost identical, they are in fact not. Fig. 3a shows incorporation of β ES into HeLa cells in complete medium, where we used high concentrations (mM) of β ES, while Supplementary Fig. 5 shows β ES incorporation in threonine-free medium, where we used low (μ M) concentrations of β ES. When comparing Fig. 3a and Supplementary Fig. 5, one can see that similar signal to noise ratio's can be attained when using μ M-concentrations of β ES threonine-free medium, as with mM-concentrations of β ES in complete medium, which we described as an increase in detection sensitivity. We have adjusted the main text to clarify the difference between the results described in Fig. 3a and Supplementary Fig. 5 (now in revised manuscript: Fig. 3a and Supplementary Fig. 4)*

..."4. No biological replicates are stated for Fig3F and it is unclear what is defined as normal and aberrant growth. Statistics should be used to define growth as aberrant."

>> *We would like to refer to our response to the first point raised by the reviewer.*

5. In the comparison between bES and HPG in HeLa cells, why was the 5 hour timepoint chosen? A typical BONCAT expt in cultured cells is 30min-60min since the media conditions are not optimal. Plus, the manuscript touts the fast kinetics of bES. A one hour comparison would be more informative to the reader in regards to proving fast kinetics as shown with BONCAT studies. The next expt is a 1hour bES treatment and the protein IDs are much lower than an expected for a BONCAT expt. If HPG incorporates better than bES, it does not take away the usefulness of bES, because bES can be used in complete media.

>> *Although metabolic protein labeling is often applied for short time intervals, it is not limited to that application. For instance, Howden et al. developed QuaNCAT, a combination of heavy isotope labeling and BONCAT, and labeled cells for 4 h to enrich newly synthesized proteins (Howden et al., QuaNCAT: quantitating proteome dynamics in primary cells, Nat. Methods, 10, 343-346, (2013)). We have tested the fast kinetics of THRONCAT by labeling Ramos cells for 60 minutes and actually found a similar number of newly synthesized proteins as Howden et al. Like in their QuaNCAT experiments, we employed a combination of heavy isotope (Lys_8) and bioorthogonal amino acid labeling. The fact that we find a relatively low number of identified proteins in the Ramos labeling experiment, compared to the HeLa labeling experiment, is possibly due to the fact that we took very stringent parameters to confidentially assign a protein as 'newly synthesized' (e.g. the peptides should contain Lys_8 and be found in all three biological replicates). Also, a shorter labeling time in Ramos cells were labeled for only 60 minutes, while HeLa cells were labeled for 5 h.*

..."6. Line 186 states THRONCAT does not enrich for Thr containing proteins, but that is exactly what it does. Confusing statement."

>> *We agree with the reviewer and adjusted the text in the manuscript to indicate that THRONCAT enriches thr-containing proteins, but does not show excessive bias towards proteins that have a high threonine content.*

..."7. The incorporation of bES in *Drosophila* is very interesting but incomplete. How well was the entire organism labeled? Were there any negative effects on the development of this complex organism? It is stated this expt was chosen because the authors previously published in *Nature Communications* the incorporation of ANL in *Drosophila*. It was disappointing that a bES and ANL comparison was not performed. In the ANL paper, the negative effects of ANL on *Drosophila* development was thoroughly investigated. It was a very well-done comprehensive study! With the negative growth effect of bES after 24hours, I would expect it would disturb normal fly development after 48 hours. I expect the same rigor as their previous *Nature Communications* paper."

>> *We thank the reviewer for the comments and performed additional experiments to address the points raised. Beyond the analysis of the effect of 48 h βES administration on larval body length, motor neuron cell body area, neuromuscular synapse length on muscle 8 and muscle 8 surface area (see above), we have also analyzed the effect on the expression pattern of the presynaptic active zone marker Brp and the postsynaptic marker Dlg1 at the larval neuromuscular junction, shown in the newly added Supplementary Figures 21. No effect on the morphology of the pre- and postsynaptic side of the NMJ could be detected. Furthermore, we evaluated whether 48 h βES incorporation would induce a developmental delay. As shown in Supplementary Fig. 22a in the revised manuscript, we found a slight delay in the time to pupa formation (median time to pupa formation increased by 12h), and the time to adult eclosion did not reveal a further developmental delay (revised manuscript: Supplementary Fig. 22b). Thus, we conclude that 48 h βES administration to larvae induces a slight developmental delay.*

We also evaluated the effect of 48 h βES administration to adult flies, by analyzing their motor performance (negative geotaxis assay). This analysis did not reveal any effect of βES administration of adult motor performance (revised manuscript: Supplementary Fig. 20f), and therefore argues against toxicity induced by βES administration to adult flies.

*We further performed a direct comparison between THRONCAT and ANL-based cell-type-specific FUNCAT in motor neurons, and included these new data in revised Fig. 5g. In *Drosophila*, the comparison between βES and HPG, as well as the newly added comparison between βES and ANL-based cell-type-specific FUNCAT clearly show that βES incorporation leads to more efficient labeling of newly synthesized proteins than HPG or ANL incorporation (revised Fig. 5f, g).*

..."8. Since this is a new compound for mass spectrometry analysis, it needs to be verified that the bES containing peptides have the expected mass shift."

>> *We have performed additional analysis to quantify the incorporation efficiency of bES compared to heavy threonine (Thr₅) as described in the first point raised by reviewer 1. We observed that βES gives the expected mass shift.*

..."9. AHA does compete with Met and is often coupled with Met starvation or depletion, but it has been successfully used without Met depletion. Please correct the text.

Liu et al. Elife 2018 Feb 7;7:e33420

Evans et al EMBO J 2019 Jul 1;38(13):e101174”

>>We have included a paragraph in the discussion and the references as proposed by the reviewer.

...” 10. Why was bES chosen over 4-FT or B-HNV?”

>> *The reviewer is correct that 4-FT or B-HNV are also excellent metabolic substrates for replacement of threonine residues as described, however, both of these compounds do not contain a bioorthogonal handle, like the alkyne present in bES, that can be used for conjugation of fluorescent dyes or affinity handles for visualizing or isolating newly synthesized proteins, respectively.*

...”11. Can bES be phosphorylated? There are many well-known thr kinases(i.e. MAPK) that are essential for cellular physiology. Please discuss in the text how you envision this could affect the application of THRONCAT.”

>> *This is an excellent point raised that we are currently looking in to. There is some data on phosphorylation data for β -HNV (Docherty, P.A., Aronson, N.N., Effect of the Threonine Analog β -Hydroxynorvaline on the Glycosylation and Secretion of α_1 -Acid Glycoprotein by Rat Hepatocytes, 260, 10847-10855, (1985)), which is structurally similar to bES, that shows that phosphorylation, although reduced, is still possible on this residue. Even if problematic for bES, we expect that only a fraction of the threonines is replaced with bES and there will always be endogenous proteins containing normal threonine present in the cell. Combined with the data on toxicity as described above, we do not expect a large impact on cell fitness.*

REVIEWER COMMENTS

Reviewer #1 (Remarks to the Author):

I appreciate the authors' efforts to address my concerns, but I still have concerns about the method used to calculate the labeling efficiency of β ES. According to the authors' description, this calculation was based on a limited set of data, specifically "paired comparison between peak intensities of more than 100 abundant β ES-/Thr5- modified peptides" of 124 peptides (presumably 62 peptide pairs?), but not on the whole detected proteome, which may introduce bias into the results. Moreover, this calculation is based on the assumption that all NSPs will be incorporated with β ES and show detectable peaks. The detection of only 124 paired peptides actually doesn't support this assumption. For instance, an extremely low labeling efficiency of 0% would make it impossible to detect any β ES-labeled peptides and find a peptide pair. Hence, the unpaired peptides are of great importance to evaluate the labeling efficiency, which was overlooked in this study.

Published protein turnover data have suggested that protein turnover in cultured cells, such as HeLa cells, is relatively fast. This is supported by evidence from a previous study (PMID: 21937730) on HeLa proteome turnover.

"Approximately 60% of HeLa proteins have 50% turnover values clustered within 5 h of the average turnover rate of ~20 h (Fig. 3, blue lines). However, if we correct for protein abundance, it takes ~24 h for 50% turnover of the total HeLa proteome, because a subset of abundant proteins has turnover values longer than the mean of ~20 h."

Therefore, during the 72-hour labeling duration used by the authors, which is much longer than pulse labeling time commonly used for turnover studies, the majority of the HeLa proteome would have turned over and been labeled with β ES. Considering this fast turnover rate, even the labeling efficiency is only ~2.5%, many more peptide pairs, not just the 62 pairs demonstrated by the authors, are expected to present. I would suggest the authors calculate the number of β ES-, Thr, and Thr5 sites to obtain a more objective calculation of the labeling efficiency. The ratio between β ES and Thr5 sites can provide a more accurate measure of incorporation efficiency. More, it is important to determine if there are any peptides that are only labeled with Thr5 and lack a corresponding β ES labeling. Such peptides would indicate a very low level of β ES incorporation.

To summarize, I recognize the novelty of this work and its potential applications for NSP studies. I support the publication of this work in Nature Communications. However, this study should not solely be considered as a proof-of-concept, as it should have practical applications in the future. If the incorporation efficiency is indeed low, the authors should honestly demonstrate this and acknowledge that this method may not be applicable for certain proteins, such as long-lived, low-abundance, and small proteins with limited Thr residues. It is important to include information about accurate labeling efficiency in the paper abstract for readers to assess the robustness of the method. I recommend that the authors avoid overstating the superiority of their method and provide a balanced evaluation of its strengths and limitations.

Reviewer #3 (Remarks to the Author):

The authors answered all my questions and I think this manuscript can be published.

Reviewer 1:

.... "I appreciate the authors' efforts to address my concerns, but I still have concerns about the method used to calculate the labeling efficiency of β ES. According to the authors' description, this calculation was based on a limited set of data, specifically "paired comparison between peak intensities of more than 100 abundant β ES-/Thr5- modified peptides" of 124 peptides (presumably 62 peptide pairs?), but not on the whole detected proteome, which may introduce bias into the results." Moreover, this calculation is based on the assumption that all NSPs will be incorporated with β ES and show detectable peaks. The detection of only 124 paired peptides actually doesn't support this assumption. For instance, an extremely low labeling efficiency of 0% would make it impossible to detect any β ES-labeled peptides and find a peptide pair. Hence, the unpaired peptides are of great importance to evaluate the labeling efficiency, which was overlooked in this study.

>>We thank the reviewer for the discussion on the calculation of the labeling efficiency of β ES. Indeed, we base our data on the paired comparison between peak intensities of 124 identified peptide pairs as we strongly believe this gives a more complete and fairer comparison on the incorporation efficiency of β ES. We directly compared peak intensities of β ES to Thr5 within the same paired peptide, because the peptide peak intensity is a good measure for abundance of the parent protein. Both β ES and Thr5 are added simultaneously to the HeLa cells and their relative abundance is measured after 72 hours, allowing for a direct comparison of relative incorporation rates between β ES and Thr5. As β ES is not incorporated as efficiently as Thr5 or Thr, there are consequently less intense peptide peaks present for β ES-containing peptides (~40-fold less); we have run our mass-spectrometers in data-dependent acquisition (DDA) mode, which has been well established to lead to underrepresentation of low-abundant peptides. As such, we expect that many β ES-containing peptides are not detected because they fall below the detection threshold of the mass spectrometer. Indeed, we see that the 124 peptides detected with a β ES modification are derived from the most abundant proteins in the dataset (based on a new analysis, see new Supplementary Fig. 7a). As seen in supplementary Figure 7, we find within these 124 peptides pairs, that the Thr5 labeled peptides were ~40x more intense than the β ES labeled peptides, which led to our conclusion that the relative incorporation rate of β ES compared to threonine is ~1/40, or ~2.5%.

.... "Published protein turnover data have suggested that protein turnover in cultured cells, such as HeLa cells, is relatively fast. This is supported by evidence from a previous study (PMID: 21937730) on HeLa proteome turnover.

"Approximately 60% of HeLa proteins have 50% turnover values clustered within 5 h of the average turnover rate of ~20 h (Fig. 3, blue lines). However, if we correct for protein abundance, it takes ~24 h for 50% turnover of the total HeLa proteome, because a subset of abundant proteins has turnover values longer than the mean of ~20 h."

Therefore, during the 72-hour labeling duration used by the authors, which is much longer than pulse labeling time commonly used for turnover studies, the majority of the HeLa proteome would have turned over and been labeled with β ES. Considering this fast turnover rate, even the labeling efficiency is only ~2.5%, many more peptide pairs, not just the 62 pairs demonstrated by the authors, are expected to present. I would suggest the authors calculate the number of β ES-, Thr, and Thr5 sites to obtain a more objective calculation of the labeling efficiency. The ratio between β ES and Thr5 sites can provide a more accurate measure of incorporation efficiency. More, it is important to determine if there are any peptides that are only labeled with Thr5 and lack a corresponding β ES labeling. Such peptides would indicate a very low level of β ES incorporation."

>>We chose to label the cells for 72 hours as we deliberately did not want to bias the data towards proteins with fast or slow turnover rates. After 72 hours labeling and without using an enrichment protocol, we detect 7410 peptides containing threonine, 9944 peptides containing Thr5 and 142 peptides containing BES (added as source data for supplementary Figure 7A)(NB. normal SILAC experiments require conditioning of the cells for 5 passages with heavy arginine to ensure full incorporation of labeled amino acids). Of the 142 BES-containing peptides, 124 of these we also found the Thr5 incorporated counterpart. Indeed, we also observe many peptides that are only labeled with Thr5 and where we do not detect the corresponding BES peptide. This is also expected as, for example, low abundant Thr5-incorporated peptides may be detected, and the BES-incorporated peptide fall below the detection threshold as they are 40x less abundant. We therefore believe that when looking at the complete proteome the less intense BES peptides fall under the threshold thereby giving a skewed picture of incorporation efficiency. Indeed, looking in more detail in the identity of the BES-containing peptides, the detected peptides all arise from the most abundant proteins. We have added a sentence in the manuscript and a rank plot figure supplementary 7A supporting this observation.

We further have to emphasize that the purpose of using BES is to label newly synthesized proteins, a subfraction of the proteome, which can be enriched from the pre-existing proteome via a pulldown procedure using azide-containing beads. The absolute incorporation efficiency of BES is of importance, but trivial because of this enrichment step.

... "To summarize, I recognize the novelty of this work and its potential applications for NSP studies. I support the publication of this work in Nature Communications. However, this study should not solely be considered as a proof-of-concept, as it should have practical applications in the future. If the incorporation efficiency is indeed low, the authors should honestly demonstrate this and acknowledge that this method may not be applicable for certain proteins, such as long-lived, low-abundance, and small proteins with limited Thr residues. It is important to include information about accurate labeling efficiency in the paper abstract for readers to assess the robustness of the method. I recommend that the authors avoid overstating the superiority of their method and provide a balanced evaluation of its strengths and limitations."

>>We thank the reviewer for supporting our work for publication in Nature Communications. We honestly feel that we give a fair comparison and we do not overstate the superiority of our method. In our manuscript, we show that BES allows for the enrichment of up to thousands of proteins (Fig. 4), showing its practical applications for subproteome profiling. We also discuss potential issues such as protein half-lives (Supplementary Fig. 14) and relevant limitations, such as decreased detection of proteins with few threonine residues (Supplementary Fig. 15). Moreover, to support the applicability of the method described in this manuscript: based on our pre-print, we have distributed BES to several researchers (>20) in the field and received impressive and supportive feedback on using THRONCAT, indicating that the incorporation of BES into proteins is efficient and the method has many practical applications.

REVIEWERS' COMMENTS

Reviewer #1 (Remarks to the Author):

I would like to thank the authors for their efforts in revising the manuscript. However, I still have concerns regarding the method used by the authors to calculate the overall labeling efficiency.

As I previously mentioned, the calculation based solely on the paired peptides introduces a significant bias. The authors' data of paired peptides clearly indicates that different sites have different incorporation rates for β ES, and a large number of unpairing of Thr5-only peptides cannot be solely attributed to their low abundance. Moreover, if the difference is only about 40 fold, current mass spectrometers are capable of handling this dynamic range for most signals.

Based on the data provided by the authors (7410 peptides containing threonine, 9944 peptides containing Thr5, and 142 peptides containing BES), a more accurate estimation of the site incorporation rate would be approximately $142/(142+9944)=1.4\%$, which is actually not significantly different from the authors' reported value of 2.5%.

I suggest that the authors clarify and discuss the difference between site and protein labeling efficiencies. While the incorporation rate at a given threonine site may be low, the labeling of an intact protein may be much better. It means that a given protein with multiple threonines could have a higher probability of synthesizing a protein molecule with at least one β ES incorporation. Using a protein with 20 Thr residues and a 1.4% incorporation rate as an example, the probability of synthesizing a protein molecule with at least one β ES incorporation would be $1-P(X=20) = 1-C(20,0) * 0.986^{20} * 0.014^0 = 1-0.754=0.246$. Therefore, ~25% of all newly synthesized protein molecules would be labeled by at least one β ES, which is relatively sufficient for applying this method in studies focusing on the protein level. Given that Thr is a high-abundance residue in the proteome, a medium-sized or larger protein would contain enough Thr residues and generate a reasonable labeling efficiency on the protein level. This discussion should provide more insights into the metabolic labeling technology.

..." As I previously mentioned, the calculation based solely on the paired peptides introduces a significant bias. The authors' data of paired peptides clearly indicates that different sites have different incorporation rates for β ES, and a large number of unpairing of Thr5-only peptides cannot be solely attributed to their low abundance. Moreover, if the difference is only about 40 fold, current mass spectrometers are capable of handling this dynamic range for most signals.

Based on the data provided by the authors (7410 peptides containing threonine, 9944 peptides containing Thr5, and 142 peptides containing β ES), a more accurate estimation of the site incorporation rate would be approximately $142/(142+9944)=1.4\%$, which is actually not significantly different from the authors' reported value of 2.5%.

I suggest that the authors clarify and discuss the difference between site and protein labeling efficiencies. While the incorporation rate at a given threonine site may be low, the labeling of an intact protein may be much better. It means that a given protein with multiple threonines could have a higher probability of synthesizing a protein molecule with at least one β ES incorporation. Using a protein with 20 Thr residues and a 1.4% incorporation rate as an example, the probability of synthesizing a protein molecule with at least one β ES incorporation would be $1-P(X=20) = 1-C(20,0) * 0.986^{20} * 0.014^0 = 1-0.754=0.246$. Therefore, $\sim 25\%$ of all newly synthesized protein molecules would be labeled by at least one β ES, which is relatively sufficient for applying this method in studies focusing on the protein level. Given that Thr is a high-abundance residue in the proteome, a medium-sized or larger protein would contain enough Thr residues and generate a reasonable labeling efficiency on the protein level. This discussion should provide more insights into the metabolic labeling technology."

... >> We thank the reviewer for the discussion on the incorporation of the labeling efficiency of β ES. We agree with the reviewer that the labeling efficiency of β ES is at least 1.4% based on the detected peptides that contain Thr5 and β ES. Though, we believe a paired peptide comparison provide a more appropriate value as we believe that some peptides may not be detected due to the detection limit of the machine. Yet, we agree with the reviewer that the calculated efficiencies are not significantly different and trivial as an enrichment step of the labeled peptides is usually performed to enhance the number of detected peptides. Yet, to provide more insight in the technology, we have added the following sentences to the manuscript:

.. "From the $\sim 10,000$ labeled peptides we found 142 peptides containing β ES indicating a minimal labeling efficiency of 1.4% (Supplementary Fig. 7a)."

And

..." Even though a few percent of all threonine sites are replaced with β ES in complete medium, we foresee that the incorporation level is sufficient to cover NSPs throughout the proteome. Indeed, after labeling HeLa cells for 5 hours, we identified more than 3000 NSPs after enrichment, which is comparable to that observed when using HPG in methionine-depleted medium (Fig. 4b and Sup. Fig 12)."